# Functional Division of Insect Blood Cells by Single-Cell RNA-Sequencing and Cell-Type-Specific FISH Markers

**DOI:** 10.3390/cells14231842

**Published:** 2025-11-22

**Authors:** Falguni Khan, Gahyeon Jin, Mojtaba Esmaeily, Shiva Haraji, Niayesh Shahmohammadi, Yonggyun Kim

**Affiliations:** Plant Medicals, School of Life Sciences & Engineering, Gyeongkuk National University, Andong 36729, Republic of Korea; falgunikhan2942@gmail.com (F.K.); gsh07129@daum.net (G.J.); mesmaeily18@gmail.com (M.E.); shiva.haraji.sh@gmail.com (S.H.)

**Keywords:** immunity, hemocyte, insect, eicosanoid, scRNA-Seq

## Abstract

Hemocytes (insect blood cells) consist of several morphological types and perform a variety of physiological processes, including immune responses. However, we do not know how many cell types are functionally differentiated in hemocytes or how they perform independent physiological processes. To address this fundamental question, we analyzed hemocyte transcripts with a single-cell RNA-sequencing (scRNA-Seq) technique. The hemocytes were collected from larvae of a lepidopteran insect, *Spodoptera exigua*, in which four different hemocyte types were morphologically recognized. scRNA-Seq discriminated 24 hemocyte clusters based on the transcripts of each cell. The clusters were separated into seven functional groups predicted from the top three highly expressed and annotated genes in each cluster: active protein synthesis (12 clusters), apoptosis (5 clusters), melanization (2 clusters), modulating cell shape (6 clusters), antimicrobial peptide production (9 clusters), calcium homeostasis (8 clusters), and cell repairing (1 cluster). Signal components of Toll/IMD immune pathways were variably expressed among the clusters. Biosynthetic genes associated with oxylipin immune mediators were specifically expressed among the clusters. Immune effectors such as melanization and apoptosis were expressed in specific hemocyte clusters. Specifically expressed genes that discriminate hemocyte types were used to develop fluorescence in situ hybridization (FISH) markers. In addition, five new hemocyte groups, which were not among the four known hemocyte types in the transcript profile, were identified and discriminated with their specific FISH markers. The hemocyte clusters underwent dynamic changes upon immune challenge. A trajectory analysis using the transcriptome suggests at least three different hemocyte differentiation pathways. These results indicate that the hemocytes of *S. exigua* are functionally highly differentiated and exhibit a dynamic transition in response to environmental changes.

## 1. Introduction

Insects possess a potent innate immune system that orchestrates both humoral and cellular defense mechanisms against pathogens [1]. Hemocytes, the insect equivalent of the vertebrate blood cells, play a pivotal role in this defense by engaging in pathogen recognition, phagocytosis, encapsulation, and melanization [2]. In addition, hemocytes can carry out multiple and diverse functions [3]. For example, similarly to macrophages in vertebrates, differentiated hemocytes play an important role in body homeostasis through tissue repair (e.g., production of extracellular matrix around differentiating tissues) and disposal of cellular waste (e.g., phagocytosis of debris from apoptotic cells) [4]. Hemocytes are also involved in the uptake and storage of lipids and nutrients [5,6].

To understand their diverse functions, hemocytes are broadly classified into five morphotypes in Lepidoptera: prohemocyte, plasmatocyte, granulocyte, oenocytoid, and spherulocyte [7,8]. Prohemocytes act as progenitor-like cells while granulocytes and plasmatocytes play crucial roles in cellular immune responses. Oenocytoids, in contrast, contribute to humoral immunity by producing prophenoloxidase (PPO), the key enzyme in the melanization cascade, which encapsulates and neutralizes pathogens upon activation [9]. Spherulocytes, although less well understood, are hypothesized to play roles in cuticle formation and macromolecule transport, potentially contributing to developmental or structural processes [10]. In *Drosophila*, only three major hemocyte types are distinguished, plasmatocyte, crystal cells, and lamellocyte, which correspond to granulocyte, oenocytoid, and plasmatocyte of Lepidoptera based on their morphology [11].

While morphology-based classification provides a useful framework, it fails to resolve the true molecular and functional diversity of hemocytes. Morphologically similar cells can vary significantly in their gene expression, developmental stage, or immune competence [12,13]. Later, hemocyte-specific antibodies provide both pan-hemocyte antibodies and specific antibodies for the different hemocyte classes [14,15]. In lepidopteran insects, lectins and monoclonal antibodies have been used to classify the hemocyte types, but the physiological functions of the marker molecules largely remain unknown [16]. Furthermore, lineage relationships among these types remain largely speculative outside model organisms like *Drosophila melanogaster*.

Single-cell RNA-sequencing (scRNA-Seq) has revolutionized our ability to deconstruct immune systems at the cellular level by capturing the transcriptome of individual cells [17]. Differentiated hemocytes exhibited cell type-specific transcriptional changes. These findings illustrate how scRNA-Seq captures immune diversification and cell lineage dynamics with high resolution. This allows for unbiased clustering of cells based on gene expression profiles, enabling the identification of distinct cell types, functional states, and differentiation trajectories [18,19,20]. Unlike bulk RNA-sequencing, which masks heterogeneity by averaging signals, scRNA-Seq reveals rare populations and dynamic immune states that may be transient or stimulus-dependent [21,22]. In *Drosophila*, scRNA-Seq has been instrumental in redefining hemocyte classification and has identified more than nine transcriptionally distinct hemocyte clusters [13,23,24], far beyond the three traditional categories. These clusters displayed unique expression signatures indicative of distinct immune functions, maturation states, or lineage commitments. Infection models further validated scRNA-Seq’s utility. Upon parasitoid wasp infection, *Drosophila* larvae rapidly produced lamellocytes, a rare cell type not found under basal conditions. Pseudo-time analysis revealed differentiation pathways from prohemocytes to lamellocytes, suggesting stimulus-dependent hematopoiesis [24]. Similar trajectory inferences have elucidated the emergence of crystal cells and terminally differentiated immune effectors [13].

Beyond *Drosophila*, scRNA-Seq has been applied to other insect species. For example, scRNA-Seq revealed discrete hemocyte clusters corresponding to granulocytes, oenocytoids, and prohemocytes, as well as novel subsets with specific transcription factor profiles in a mosquito *Anopheles gambiae* [25]. This study identified Lozenge, a transcription factor essential for crystal cell fate in *Drosophila*, as a key regulator of oenocytoid differentiation in mosquitoes, highlighting a conserved genetic program in insect hematopoiesis. The scRNA-Seq has also provided new insights into hemocyte diversity in Lepidoptera. In *Bombyx mori*, scRNA-Seq revealed 13 distinct hemocyte clusters under basal conditions, which correspond to traditional cell types like plasmatocytes, granulocytes, and oenocytoids, while also uncovering substantial molecular heterogeneity within these categories [26]. Upon baculovirus infection, the hemocyte repertoire expanded to 20 clusters, including infection-specific populations enriched in proliferative, progenitor-like cells, suggesting hematopoietic activation in response to immune challenge [8]. Pseudo-time analyses have enabled reconstruction of developmental trajectories from prohemocytes to mature immune cells, while newly identified marker genes support precise classification and future functional studies [12,17]. Importantly, scRNA-Seq advances hemocyte research in non-model insects by providing molecular tools for identifying and characterizing immune subsets, even in species with limited genetic resources. Despite these insights, comprehensive scRNA-Seq studies in other agriculturally important moths are lacking.

The beet armyworm, *Spodoptera exigua*, is a major lepidopteran pest species with growing significance in both applied entomology and immune ecology [27]. Despite the ecological and economic importance, prior works in *S. exigua* have been based on hemocyte morphology and immune function using bulk transcriptomic or histological approaches, but no single-cell atlas of hemocytes has yet been published [28,29]. This study aimed to re-classify the hemocytes of *S. exigua* using scRNA-Seq. To predict the physiological functions of different hemocyte types, their transcripts were comparatively analyzed using differentially expressed genes (DEGs). By analyzing DEGs uniquely expressed in each cell type, molecular markers specific to different cell types were developed using a fluorescence in situ hybridization (FISH) technique. In addition, this study proposes five novel hemocyte types, which are not included among the traditional four hemocyte types detected by FISH probes. Our findings shed light on conserved and species-specific features of insect immunity and offer a valuable framework for future functional studies in lepidopteran host–pathogen interactions.

## 2. Materials and Methods

### 2.1. Insect Rearing and Immune Challenge

Larvae of *S. exigua* were reared on an artificial diet [30] under controlled conditions (25 ± 2 °C, 60 ± 5% RH, 16:8 h light:dark), progressing through five larval instars (L1–L5). Adults were supplied with a 10% sucrose solution. Newly molted L5 larvae were immune-challenged by injecting heat-killed *Escherichia coli* TOP10 (1.4 × 10^5^ cells per larva) into a proleg while naive larvae received phosphate-buffered saline (PBS, pH 7.4). The treated larvae were maintained at room temperature for 8 h prior to sampling.

### 2.2. Chemicals

Diethyl pyrocarbonate (DEPC)-treated water was prepared by adding DEPC to deionized distilled water to a concentration of 0.1% and then autoclaving it. Phosphate-buffered saline (PBS) was prepared with 100 mM Na_2_HPO_4_·12H_2_O, 18 mM KH_2_PO_4_, 28 mM KCl, and 138 mM NaCl at pH 7.4. A 20× saline-sodium citrate (SSC) buffer was prepared with 3.0 M NaCl and 0.3 M sodium citrate at pH 7.0. Yeast tRNA was purchased from Thermo Fisher Scientific (Waltham, MA, USA). Lauryl sulfate sodium salt for SDS preparation, as well as dextran sulfate and deionized formamide, were purchased from Sigma-Aldrich Korea (Seoul, Republic of Korea). Anticoagulant buffer (ACB) was prepared with 186 mM NaCl, 17 mM Na_2_EDTA, and 41 mM citric acid.

### 2.3. Hemocyte Collection for scRNA-Seq

Preparation of single hemocytes in suspension was followed by the protocol of Feng et al. [8]. Briefly, three mL of hemolymph was collected from challenged and control larvae on ice and centrifuged at 500 rpm for 2 min at 4 °C. The hemocyte pellet was washed twice using one mL of cold TC-100 medium (Welgene, Gyeongsan, Republic of Korea) supplemented with 10% heat-inactivated (55 °C for 30 min) fetal bovine serum (Welgene), which was followed by filtering through a 40 μm cell strainer (Biosharp, Hefei, China) and centrifuged at 500 rpm for 2 min at 4 °C. The hemocyte pellet was resuspended using 1× filtered PBS supplemented with 0.04% bovine serum albumin (BSA) (Sigma Aldrich Korea). The number of hemocytes was counted using a hemocytometer. Cell viabilities were assessed by the intensity of the blue signal stained by 0.4% trypan blue. Cell suspension samples with viability (>70%) and quantity (>5 × 10^5^) were subjected to single-cell encapsulation. Hemocyte classification based on the morphological criteria followed the method of Strand [2].

### 2.4. scRNA-Seq

Single-cell encapsulation, cDNA synthesis, RNA-Seq library construction, and primary data processing were performed by Macrogen Inc. (Seoul, Republic of Korea). Cells were partitioned into gel bead-in-emulsions (GEMs) on the 10× Genomics Chromium Controller using Single Cell 3′ Chemistry v2 kit (10× Genomics, Pleasanton, CA, USA) following the manufacturer’s protocol. Within each GEM, a barcoded gel bead bearing a poly(dT) primer captured poly-adenylated mRNA, yielding cell-indexed, unique molecular identifier (UMI)-tagged cDNA anchored at the 3′ end. Libraries were prepared according to the Chromium Single Cell 3′ workflow and sequenced on an Illumina NovaSeq 6000 with the paired end reads. The scRNA-Seq data were deposited to GenBank with accession numbers of GSE308201 for functional division of *S. exigua* hemocytes in immunity, GSM9240195 for scRNA-Seq of *S. exigua* hemocytes from naïve larvae, and GSM9240196 for scRNA-Seq of *S. exigua* hemocytes from immune-challenged larvae.

### 2.5. Bioinformatics

Raw FASTQ files were first evaluated with FastQC to assess read quality, adapter content, and base-level Phred scores. Reads with low base quality (Phred < 20), particularly in the cell barcode or UMI regions, were discarded. Library demultiplexing, alignment, UMI counting, and initial filtering were performed using Cell Ranger (10× Genomics, v3.1.0). UMIs differing by one base were collapsed using Cell Ranger’s built-in error-correction algorithm, and barcodes not matching or correctable to the 10× whitelist were excluded. The cleaned reads were then aligned to the *S. exigua* transcriptome using either the de novo assembly (generated with Trinity and TransDecoder v3.0.1 (https://github.com/TransDecoder/TransDecoder/wiki, accessed on 30 July 2025)) or, when applicable, the reference genome using (https://github.com/alexdobin/STAR, accessed on 30 July 2025) or Bowtie2 version 2.4.5 (http://bowtie-bio.sourceforge.net/bowtie2, accessed on 30 July 2025). Following alignment, cell-level quality control was performed, and cells were retained only if they expressed at least 200 genes and contained less than 10% mitochondrial transcripts. Cells with UMI counts below the empty-droplet threshold or with abnormally high gene numbers (>5000–6000), which typically indicate doublets, were also excluded. Genes detected in fewer than three cells were removed to eliminate technical noise. After filtering, the high-quality gene-barcode matrices were imported into Seurat v4.0 in R (https://satijalab.org/seurat/articles/get_started.html, accessed on 30 July 2025) for downstream analysis. Data were log-normalized, and the top 5000 highly variable genes were identified using the variance-stabilizing transformation (VST) method. Dimensionality reduction was performed with principal component analysis (PCA), followed by clustering using the Louvain algorithm. Cluster visualization was conducted using uniform manifold approximation and projection (UMAP) embedding.

### 2.6. DEG Analysis

To identify DEGs in hemocytes between the two treatments, naive and immune challenge, the abundances of unique genes across samples were estimated into read counts as an expression measure using the RSEM algorithm (RSEM version v1.2.29, bowtie 1.1.2, http://deweylab.github.io/RSEM/, accessed on 30 July 2025). Genes were considered differentially expressed if they had a log2 fold change > 0.25, a percentage of cells in a specific cluster where the gene is detected >25%, and an adjusted *p*-value < 0.05. For functional annotation, DEGs were annotated against multiple reference databases, including NCBI Nucleotide (GenBank/NT, Release 268.0, https://www.ncbi.nlm.nih.gov/nucleotide/, accessed on 30 July 2025), Pfam v37.1 (https://pfam.xfam.org/, accessed on 30 July 2025), Gene Ontology (GO, http://www.geneontology.org/, accessed on 30 July 2025), NCBI non-redundant Protein (ClusteredNR, https://www.ncbi.nlm.nih.gov/protein/, accessed on 30 July 2025), UniProtKB Release 2025-03 (http://www.uniprot.org/, accessed on 30 July 2025), and EggNOG v6.0.5 (http://eggnogdb.embl.de/, accessed on 30 July 2025). Searches were performed using NCBI BLASTN and DIAMOND v2.1.12 (https://github.com/bbuchfink/diamond, accessed on 30 July 2025) with a default E-value cutoff of 1 × 10^5^. GO enrichment analysis was carried out with Blast2GO v6.0. For each cluster, the three most highly expressed DEGs were selected as candidate marker genes. Their expression patterns across clusters were visualized using dot plots, where dot color indicated average expression (FPKM: Fragments Per Kilobase of exon per Million mapped reads) and dot size reflected the proportion of cells expressing the gene. All analyses were conducted with default parameters unless otherwise specified.

### 2.7. Trajectory Analysis

Trajectory inference and pseudo-time analysis were performed using Monocle 3, enabling the reconstruction of differentiation lineages among hemocyte subpopulations. For trajectory inference and lineage staging, we constructed a Monocle 3 Cell Data Set (CDS) from normalized counts and metadata, reduced dimensionality (preprocess CDS, num. dim = 50), learned a principal graph on the UMAP manifold, and ordered cells in pseudo-time. Progenitor cells, identified by canonical markers and central UMAP position, were set as the root state; the graph was minimally pruned to remove short, low-support branches. Terminal tips of the graph were clustered using k-means clustering (k = 3) to define three macro-lineages. Cells were mapped to the nearest lineage backbone by geodesic distance, and clusters were assigned to the lineage containing the majority of their cells (a majority vote). Within each lineage, cluster median pseudo-time was partitioned into tertiles to designate early (root-proximal, progenitor-enriched), mid (transitional along branches), and late (terminal, branch-end) stages.

### 2.8. Florescence In Situ Hybridization (FISH)

FISH was performed on hemocytes collected from L5 larvae of *S. exigua*. Approximately 150 μL of total hemolymph was extracted from five larvae through the prolegs and mixed with 350 μL of ACB. The mixture was placed on ice for 20 min to stabilize hemocytes and minimize spontaneous spreading. After centrifugation at 1000 rpm for 2 min at 4 °C, 250 μL of the supernatant was removed, and the remaining hemocyte suspension was gently mixed with 250 μL of TC100 insect tissue culture medium (Welgene). A 20 μL aliquot of the hemocyte suspension was placed onto a slide glass in a wet chamber under dark conditions. The cells were fixed with 4% paraformaldehyde for 10 min at room temperature (RT) and washed three times with PBS. Permeabilization was carried out with 0.2% Triton X-100 in PBS for 10 min, followed by washing with PBS. The hemocytes were then rinsed in 2× SSC and incubated at 42 °C with 25 μL of pre-hybridization buffer (2 μL of yeast tRNA, 2 μL of 20× SSC, 4 μL of dextran sulfate, 2.5 μL of 10% SDS, and 14.5 μL of deionized H_2_O) in a dark, humid chamber for 1 h. After pre-hybridization, the buffer was replaced with hybridization buffer (5 μL of deionized formamide and 1 μL of oligonucleotide probe in 19 μL of pre-hybridization buffer). DNA oligonucleotide probes were labeled at the 5′ end and purified by high-performance liquid chromatography (Bioneer, Daejeon, Republic of Korea). To identify hemocyte types, antisense probes complementary to the target mRNAs were used, with corresponding sense probes serving as negative controls (Appendix A). Slides were covered with RNase-free coverslips and incubated overnight (16 h) in a humid chamber at 42 °C. After hybridization, hemocytes were washed twice with 4× SSC for 10 min each and incubated in 4× SSC containing 1% Triton X-100 at RT for 5 min. This was followed by two washes with 4× SSC for 10 min each and one wash with 2× SSC. The samples were air-dried, and a drop of PBS/glycerol (1:1, *v*/*v*) was added before incubation at RT for 15 min. Finally, samples were mounted with a coverslip and observed under a fluorescence microscope (DM2500; Leica, Wetzlar, Germany) at ×200 magnification.

### 2.9. Immune Pathway Detection with Associated Genes

Six different immune-associated pathways (Toll/IMD immune signal, antimicrobial peptide (AMP), oxylipin biosynthetic pathway, PGE_2_ signal pathway, PPO activation, and apoptosis) were analyzed. Associated GenBank IDs were assigned for each gene from *S. exigua*, and their expressions were compared among the 24 hemocyte clusters using a heatmap with statistical support. Pathway diagrams were created using the short form of each gene name in Microsoft Office 365 (Power Point), and the heatmaps were drawn using GraphPad Prism 10.0.1 (GraphPad Software, Boston, MA, USA).

### 2.10. Statistical Analysis

Statistical analyses, including the chi-square test, ANOVA and LSD mean comparison test, were performed using the SAS 9.4 statistical analysis program. The representation of statistical figures such as DEGs and GO pathways, was created by inputting values into SigmaPlot 10.0. The heatmap categorization of immune pathway, stage-specific immune gene division, and GO analysis data were represented using GraphPad Prism 10.0.1.

## 3. Results

### 3.1. scRNA-Seq Reveals 24 Different Hemocyte Types of S. exigua

Hemocytes were collected separately from naïve or immune-challenged (‘Challenge’) larvae and subjected to scRNA-Seq (Figure 1). More than 12,500 hemocytes were assessed in each treatment, in which hemocytes derived from naïve and immune-challenged larvae were read over 36 Gb and contained 3339 and 5278 transcripts per cell on average, respectively. The resulting amplified transcripts were annotated to similar numbers of the total annotated genes in both treatments though a greater number of genes was counted per hemocyte in challenges larvae than that of naïve larvae (Figure 1a). Based on these annotated transcripts, the hemocytes were separated by principal component analysis, and the resulting separate groups were depicted by UMAP (Figure 1b). A total of 24 clusters (cluster 0 to cluster 23) were separated into five large aggregates and two small aggregates (cluster 20 and cluster 21) in *S. exigua*.

### 3.2. Immune Challenge Alters Hemocyte Types Classified by Differentially Expressed Transcripts Derived from scRNA-Seq

Immune challenge altered the hemocyte types revealed by scRNA-Seq in *S. exigua* (Figure 2). Control naïve larvae used PBS-injection instead of the bacterial injection for the treatment. Even though the overall groupings of the hemocytes were similar between the control and the treatment, specific clusters either mostly disappeared (thin arrows) or newly appeared (thick arrows) upon the immune challenge (Figure 2a). In each cluster, the influence of immune challenge was evaluated on the hemocyte cell numbers (Figure 2b). Most cell clusters showed relatively similar numbers between the naïve and immune-challenged treatments. Upon immune challenge, however, hemocyte numbers markedly increased in clusters 12 and 18, while they decreased in clusters 0, 11, 13, 15, 17, and 19.

The change in hemocyte types after the immune challenge was further analyzed by assessing the relative proportions of DEGs and uniquely expressed genes in each cluster (Figure 2c and Appendix A). As expected, clusters 17, 18, and 22, which nearly disappeared or newly appeared upon immune challenge, showed relatively high proportions of uniquely expressed genes compared to other clusters. In contrast, clusters 6 and 7 did not show any change in cell density (Figure 2b) upon immune challenge but had relatively high proportions of unique genes and DEGs (Figure 2c).

### 3.3. Seven Different Functional Hemocyte Groups Classified by Highly Expressed DEGs

Our scRNA-Seq of *S. exigua* hemocytes predicted 12,578 genes in control and 12,354 genes in immune challenge. However, these predicted gene numbers varied from 3811 (cluster 6) to 12,059 (cluster 0) genes among 24 hemocyte clusters (Appendix A). Although a few genes were specific to each hemocyte cluster, 234 genes were uniquely expressed in cluster 19. To classify the hemocyte clusters based on the DEGs, the highly expressed genes were selected for each cluster (Figure 3, Appendix A). Among the 24 clusters, 12 clusters (0, 3, 7, 8, 10, 14, 15, 16, 17, 19, 21, and 22) are likely to be active in protein synthesis in either naive or immune-challenged treatments because they showed high expressions of ribosomal proteins or translation factors. Five clusters (1, 6, 9, 18, and 23) highly expressed apoptosis-associated genes. Two clusters (4 and 13) are likely associated with melanization because they highly expressed *PPO* or *paired mesoderm homeobox 2A*, which regulates genes involved in catecholamine synthesis, like tyrosine hydroxylase and dopamine β-hydroxylase, likely for melanin formation in the hemocytes [31]. Six clusters (2, 7, 11, 20, 21, and 23) are likely to be undergoing changes in cell shape upon immune challenge due to the high expression of genes like *secreted protein acidic and cysteine-rich* (*SPARC*), *hemicentin*, or *myosin*. SPARC is a cysteine-rich, acidic, matrix-associated protein involved in collagen calcification in bone for vertebrates and also in extracellular matrix synthesis to promote changes in cell shape [32]. Hemicentin is a very large, ancient protein that belongs to the fibulin family of extracellular matrix components and is essential for guiding cell migration [33]. Nine clusters (3, 8, 12, 14, 15, 16, 17, 20, and 21) may be responsible for AMP production, such as cecropins and spodoptericin. Eight clusters (2, 7, 8, 9, 15, 18, 19, and 22) are likely to be associated with calcium homeostasis because they highly expressed a sodium/calcium exchanger regulatory protein in naïve or immune-challenged treatments. Hemocytes in Cluster 5 are likely associated with cell-repairing because they highly expressed *slowmo* (a gene whose protein is localized in mitochondria) and mesencephalic astrocyte-derived neurotrophic factor homolog (a gene whose protein plays a role in protection and repair in response to the unfolded protein response or endoplasmic reticulum stress) [34] in both naïve and immune-challenged conditions. This analysis suggests that the hemocytes are classified into seven functional groups with their characteristic gene expressions: active protein synthesis, apoptosis, melanization, modulating cell shape, AMP production, calcium homeostasis, and cell-repairing.

### 3.4. Hemocyte Types Associated with Immunity of S. exigua

Toll/IMD immune signaling pathways were analyzed using scRNA-Seq (Figure 4a). In the Toll pathway, small peptidoglycan recognition proteins (PGRPs) and β-1,3-glycan receptor were expressed in most hemocyte clusters. Among the four spätzle genes, *spätzle 1* was expressed in most hemocyte clusters compared to other three spätzle genes. Among the four Toll receptors, *Toll-4B* was dominantly expressed in most hemocyte clusters while *Toll-10* was expressed only in cluster 10. Subsequent Toll adaptors (*MyD88* and *Pelle*) were expressed in most hemocyte clusters. Finally, nuclear receptors (two *Dorsal* genes) and their inhibitory regulators (IκBs) were also expressed in most hemocyte clusters. In IMD pathway, large PGRPs were expressed in most hemocyte clusters except clusters 5, 6, and 23. Subsequent signal components from IMD to Relish were also expressed in most hemocyte clusters. These Toll/IMD immune signal pathways led to expressions of various AMP genes like *lysozymes*, *cecropins*, *attacin*, *moricin*, and *spodoptericin* from the hemocytes.

Genes associated with oxylipin biosynthesis were analyzed in different hemocyte clusters (Figure 4b). Three phospholipase A_2_ (PLA_2_) genes along with fatty acid elongase/desaturase were expressed in specific cell clusters. Compared to the two *iPLA_2_* genes, *sPLA_2_* was expressed in a greater number of hemocyte clusters. In two *iPLA_2_* genes, *iPLA_2_-B* was more frequently expressed in the hemocyte clusters than *iPLA_2_-A*. Two prostaglandin (PG) synthetic genes (*PGES* and *PGIS*) for PGE_2_ and PGI_2_ production were expressed in most hemocyte clusters except clusters, 4 and 23. Genes associated with PG metabolism (*PG reductase* and *PG dehydrogenase*) were also expressed in specific hemocyte clusters, in which cluster 22 highly expressed *PG reductase* at the immune challenge. At least one of the four epoxyeicosatrienoic acid (EET) synthetic genes (*EPX1*–*EPX4*) was expressed in most hemocyte clusters except cluster 23, in which *EPX1* and *EPX2* were dominantly expressed in the hemocytes. Genes associated with leukotriene (LT) biosynthesis were also expressed in different hemocyte clusters, in which the gene (*LTA_4_H*) of LTA_4_ hydrolase converting LTA_4_ into LTB_4_ was highly expressed in most hemocyte clusters. Finally, another oxylipin, epoxyoctadecamonoenoic acid (EpOME), plays a crucial role in immunity control as a negative regulator in *S. exigua* [35]. The EpOME synthase gene was also expressed in specific hemocyte clusters.

PGE_2_ signal transduction in hemocytes was analyzed by its signal component expressions using scRNA-Seq (Figure 4c). The *PGE_2_ receptor* (*PGE_2_R*) was expressed in specific hemocyte clusters while three different *Gα* subunits in the trimeric G proteins associated with G protein-coupled receptors (GPCRs) were expressed in most clusters. *Adenylate cyclase* (*AC*) was expressed in specific hemocyte clusters (7, 14, and 20–22). The *IP3 receptor* (*IP3R*) was expressed in most hemocyte clusters except clusters 6 and 23. *Ryanodine receptor* (*RyR*) and *aquaporin* (*AQP*) were expressed in specific hemocyte clusters while *protein kinase A* (*PKA*) and *F-actin* were expressed in all hemocyte clusters. These analyses indicate that although the hemocytes of *S. exigua* possess the cellular components necessary for PGE_2_ signal transduction—from ligand binding to the activation of secondary messengers such as Ca^2+^ and cAMP—only specific hemocyte clusters respond to PGE_2_.

Melanization (Figure 4d) and apoptosis (Figure 4e), which are two major immune effectors, were assessed by their component gene expressions. For melanization, phenoloxidase (PO) needs to be activated from PPO by a proteolytic cleavage of serine protease homolog (SPH) and PPO-activating protease (PAP) [38]. Two *PPO* genes, four *PAP* genes, and one *SPH* gene were expressed in specific hemocyte clusters, in which two PPO genes were highly expressed in cluster 19 at naïve and cluster 13 at immune-challenge. Four serpin genes (*SPN1*-*SPN4*) were expressed in different hemocyte clusters, in which *SPN1* and *SPN2* were dominantly expressed in the hemocytes. For apoptosis, five caspase genes expressed the hemocytes among eight caspase (*Cas-1*~*Cas-8*) genes, in which two caspase genes (*Cas-2* and *Cas-5*) were dominantly expressed. Although three genes (*MAPK*, *Cyt C*, and *APAF*) responsible for inducing apoptotic signal were expressed in most hemocyte clusters but the apoptosis may be restricted to specific hemocyte clusters because *Cas-1*, a final apoptotic executor, was expressed at specific hemocyte clusters. It is noticeable that *inhibitor of apoptosis* (*IAP*) was expressed in the most hemocyte clusters. These analyses indicate that the specific hemocytes perform melanization and apoptosis to defend against microbial infection.

### 3.5. Hemocyte Classification Using Cell Type-Specific Markers

Four hemocyte types are classified in *S. exigua* hemolymph with morphological characters, similar to another lepidopteran insect, *Pseudoplusia includens* [39,40]. Feng et al. [26] selected marker genes specific to hemocyte types of *B. mori*: granulocyte (GR), plasmatocyte (PL), oenocytoid (OE), and spherulocyte (SP). We tested whether these marker genes were suitable to classify different hemocyte clusters of *S. exigua* (Appendix A). Five PL candidate markers were highly expressed in clusters 4 and 5, in which *paired mesoderm homeobox protein 2A-like* (*PMH*) was the most highly expressed and selected for subsequent FISH analysis. Twelve GR candidate markers were expressed in at least 11 different clusters in naïve or challenged larvae, in which *cecropin-B1* was the most highly expressed and selected for subsequent FISH analysis. Two OE candidate markers were expressed in specific clusters, in which *PPO-2* was the most highly expressed and selected for subsequent FISH analysis. Four SP candidate markers were expressed only in cluster 22, in which *Repat9* was the most highly expressed and selected for subsequent FISH analysis.

The selected four marker genes (*PMH*, *CecB1*, *PPO2*, and *Repat9*) discriminating four different hemocyte types were compared in their expressions in the 24 hemocyte clusters between control and immune-challenge treatments (Figure 5a). Most marker genes were specifically expressed in different clusters except cluster 22 expressed both *CecB1* and *Repat9*. Some clusters such as 1, 5, 6, and 9 expressed these markers at low levels in both naïve and immune challenge. Then, we applied these marker genes to hemocytes of *S. exigua* with the FISH technique (Figure 5b). The marker genes appeared to stain the specific hemocyte types when the positive cells were compared to the hemocyte types determined by morphological characters (Appendix A). When the hemocyte frequencies classified by the morphological characters were compared to the positive frequencies of different FISH probes (Figure 5c), they were not very significantly different (X^2^ = 6.90; df = 3; *p* = 0.075), although there was a marked decrease (>20%) in GR frequency in FISH analysis.

Based on this validation of the marker genes, the 24 hemocyte clusters were classified into five hemocyte types: 12 GRs, 2 PLs, 4 OEs, 1 SP, and 5 novel hemocyte types (nHC1–nHC5) (Figure 6a). The classification of nHC1–nHC5 was based on three complementary criteria: (i) their distinct transcriptional signatures, including uniquely expressed marker genes identified from cluster-specific DEG analyses (Appendix A), (ii) their clear separation from classical hemocyte clusters in UMAP space, and (iii) their specific hybridization patterns with FISH probes that did not overlap with those used for GR, PL, OE, or SP. These combined transcriptomic and cytological features support their designation as distinct subpopulations. For further clarification of these nHCs, five specific marker genes were selected from the uniquely expressed genes in each hemocyte cluster (Appendix A). The resulting five FISH probes could specifically detect different nHCs (Figure 6b). These nHC types showed 3–4% relative proportions among the total hemocytes and summed up to almost 16%. Based on the FISH classification, the five hemocyte types (GR, PL, OE, SP, and nHC) were depicted on the UMAPs drawn from scRNA-Seq (Figure 6c). The immune challenge clearly changed the relative distribution of these hemocyte types.

To explain the dynamic change in the hemocyte types, the clusters obtained from scRNA-Seq were monitored by a trajectory analysis using the change in the gene expressions (Figure 7). Pseudo-time analysis of integrated hemocyte clusters using Monocle 3 resolved the data into three major lineages. Prohemocytes were defined as the root state, representing the earliest progenitor cells. Each lineage progressed along a continuous trajectory that could be subdivided into three developmental stages: an early stage, dominated by progenitor-like clusters at the root; a mid-stage, composed of transitional clusters located along branching paths; and a late stage, containing terminally differentiated hemocyte subtypes at the branch termini. By mapping cells to these trajectories, we captured the stepwise progression of hemocyte differentiation and identified both conserved and immune challenge–specific shifts in lineage commitment. Clusters were assigned to lineages based on the predominant cell type represented within each cluster. Clusters C20– C23 comprised only a small number of cells; consequently, they did not receive robust lineage labels during principal-graph mapping and were initially unassigned. This yielded coherent, directional progressions rather than a single, global path, indicating multiple differentiation routes across the dataset. Lineage 1 progressed from a plasmatocyte (PL1) to a granulocyte (GR4) via novel type of hemocytes: C4 → C9 → C10 → C2 → C6 → C8. Lineage 2 followed the progression route from a granulocyte (GR7) to a novel type of hemocyte (nHC4): C7 → C17 → C14 → C3 → C5 → C15 → C16 → C18. Lineage 3 had a progressive order from oenocytoid (OE3) to a novel hemocyte (nHC3): C13 → C0 → C12 → C1 → C11 → C19.

## 4. Discussion

Unlike vertebrates, the insect open circulatory system is responsible for the delivery of nutrients, metabolites, and hormones, in addition to providing immunity, as it is not involved in oxygen delivery. Hemocytes within this open circulatory system perform both immune and other physiological roles. Thus, insects and other invertebrates possess different types of hemocytes, each with apparently discrete physiological roles. However, the unique roles of these different hemocyte types remained unclear until the advent of transcriptome analysis using scRNA-Seq. This study investigated the different hemocyte types of *S. exigua* based on their transcripts and separated them into 24 cluster types. These clusters differed in the kinds of highly expressed genes and exhibited variation in their expression profiles of immune-associated genes. With the exception of five clusters, most of the 24 clusters were assigned molecular markers using specific genes expressed in each of the four traditional hemocyte types. The 24 hemocyte clusters were not static but exhibited dynamic changes in their clustering patterns upon immune challenge.

The hemocyte types of *S. exigua* are further divided from four morphological types into 24 transcriptomic clusters by scRNA-Seq. The 24 clusters were distinct in their major transcripts (top three highly expressed genes) and were classified into seven physiological functions categories. Interestingly, five clusters (1, 6, 9, 18, and 23) highly expressed the apoptosis-associated genes and were identified as five novel hemocyte types (nHC1-5) based on molecular markers. Apoptosis plays a crucial role in insect cellular immune responses, including nodule formation or encapsulation [41]. In addition, hemocyte apoptosis can trigger the proliferation of neighboring cells in *Drosophila* [42]. This suggests that the nHCs undergo apoptosis to directly defend against pathogens and also indirectly repair neighboring hemocytes or other tissues damaged from infection. This needs further analysis to fully understand the physiological functions of these new hemocytes. Cluster 13 highly expressed PPO, presumably to produce melanin, and was identified as an oenocytoid (OE3) with its specific molecular marker. The other three oenocytoids highly expressed genes associated with protein synthesis or calcium homeostasis (OE1 and OE4) or changes in cell shape (OE2). This suggests that oenocytoids perform various physiological processes other than melanization. For example, oenocytoids in *Helicoverpa armigera* produce a cytokine and release it upon microbial infection to activate plasmatocytes [43].

Our scRNA-Seq reveals that PGE_2_ signaling is restricted to specific hemocyte cluster types, although eicosanoid biosynthesis occurs in most hemocyte types of *S. exigua*. Eicosanoids, which are oxygenated polyunsaturated fatty acids, mediate various physiological functions in insects, including metabolism, secretion, reproduction, and immunity [44]. They are produced from a C20 precursor, usually arachidonic acid, and are classified into PGs, LTs, and EETs. Our current scRNA-Seq showed that the hemocytes express the genes associated with the biosynthesis of these three eicosanoids. Although insects do not encode the vertebrate lipoxygenase (LOX) orthologous gene, we did not find a LOX transcript for LT production. However, our scRNA-Seq predicts the expression of LTA_4_ hydrolase, which converts LTA_4_ to LTB_4_, in the hemocytes. Scarparti et al. [45] predicted LTA_4_ synthase from *Drosophila* genome. Both LTA_4_ and LTB_4_ are functional in the cellular immunity of *S. exigua* [46,47]. In PG production, the hemocyte transcriptomes showed that three different *PLA_2_* genes, cyclooxygenase-like genes (*POX-F* and *POX-H* [48]), and PGE_2_/PGI_2_ synthase genes are expressed. For EET production, four epoxygenase genes are expressed in the hemocytes, suggesting the production of all four EETs. These findings suggest that the hemocytes of *S. exigua* produce these eicosanoids, at least for immune responses. In addition, the hemocytes expressed the components for EpOME biosynthesis and metabolism. EpOME acts as an insect resolvin to suppress excessive and unnecessary immune responses, typically during late infection stage [49,50]. Eleven hemocyte clusters expressed the EpOME synthase, in which four clusters (5, 8, 19, and 22) expressed it only in hemocytes from naïve larvae. This suggests that EpOME may have additional physiological role(s) beyond that of an immune suppressor.

PGE_2_ is a main eicosanoid in insects, mediating various immunological functions [44]. However, the PGE_2_ receptor was expressed in only nine clusters, in which three of them (3, 8, and 10) expressing it without any immune challenge were granulocytes (GR2, GR4, and GR5). In contrast, cluster 21 expressed it only upon immune challenge and was another type of granulocyte (GR12), while five constitutively expressed clusters (0, 4, 5, 7, and 19) included oenocytoids (OE1 and OE4), plasmatocytes (PL1 and PL2), and a granulocyte (GR3). PPO is an inactive enzyme produced specifically from oenocytoids in *S. exigua* and is released via cell lysis into the plasma, where it undergoes a proteolytic cleavage to become its active form [51]. This oenocytoid cell lysis is triggered by PGE_2_ via its receptor [52]. During cellular immune responses, PGE_2_ activates hemocyte spreading, phagocytosis, and nodulation of plasmatocytes and granulocytes [53]. These functions help explain the PGE_2_ receptor gene expression in these hemocyte clusters during immune challenge. In contrast, the three clusters that express the PGE_2_ receptor only under naïve conditions suggest alternative functions of these granulocytes. Notably, these three clusters are classified into ‘active in protein synthesis’ within the six functional categories. They may be undergoing cell differentiation of the granulocytes during development, such as GR4 in our hemocyte trajectory analysis in lineage 1. For example, PGE_2_ alone can initiate a hemocyte differentiation from a stationary to a circulatory type in *S. exigua*, where plasmatocyte density is significantly enhanced for subsequent differentiation into granulocytes [40]. The differentiation from plasmatocytes to granulocytes is also predicted from our trajectory analysis in lineage 1. In addition to PGE_2_, the Toll signal is known to be associated with hemocyte differentiation in a mosquito [54]. Our scRNA-Seq showed four Toll receptors are expressed in the hemocytes of *S. exigua* with or without immune challenge. This kind of hemocyte differentiation is called trans-differentiation, in which a mature cell type transforms into another mature cell type, distinct from hemocyte differentiation at the hematopoietic organ [55]. These findings suggest that hemocytes differentiate during development or upon immune challenge, exhibiting their functional plasticity.

The hemocyte differentiation of *S. exigua* was predicted using a trajectory analysis and showed at least three lineages: plasmatocyte (PL1)–granulocyte (GR4), granulocyte (GR7)–novel hemocyte (nHC4), and oenocytoid (OE3)–novel hemocyte (nHC3). In *Drosophila*, the larval hemocytes basically originate from the procephalic and cardiogenic mesodermal hematopoietic organs [56]. The cardiogenic hematopoietic organs, called lymph glands, are situated on each side of the dorsal vessel. The paired primary lobes of the lymph glands consist of a medullary zone with progenitor cells, a cortical zone with differentiated hemocytes, and a posterior signaling center that supervises the maintenance and differentiation of progenitor cells [57]. The procephalic origin gives rise to the peripheral and sessile hemocytes attached to the body wall until being released into the circulatory hemolymph [58]. In Lepidoptera, hemocytes originate from the mesodermal stem cells in the hematopoietic organ located in the thoracic segments near the anterior and posterior wing discs [59] and differentiate into the mature morphological types when they leave the organ. This is because most hemocytes in the organ are prohemocytes (60–70%) and oenocytoids (30–40%), while granulocytes are much less frequent cells (less than 1%) [60]. Hemocyte differentiation may also occur in the hemolymph because the progenitor cells (prohemocytes and oenocytoids) transform in a short time (10–30 min) when incubated in vitro [60]. This is a clearly contrasting strategy for the hemocyte differentiation between *Drosophila* and lepidopteran insects. It also supports the dynamic change in the hemocyte cluster types in *S. exigua* found in our current study.

This study developed molecular markers to discriminate classical hemocyte types based on genes uniquely expressed in each cell type. The molecular FISH probes specifically reacted to each hemocyte type. In addition, this study also developed FISH probes to detect five novel hemocyte types (nHC1-nHC5), which were not distinct from GRs with morphological characters. A similar strategy using monoclonal antibodies to discriminate with molecular probes was conducted in *P. includens* [61]. The monoclonal antibody successfully discriminated the different hemocytes but showed some cross-reactivity with insect development or after immune challenge, suggesting the trans-differentiation observed in our current study. Compared to the monoclonal antibody markers, our FISH probes can more specifically discriminate target cells based on sequence complementarity. A FISH probe targeting *cecropin B1* mRNA was developed to recognize granulocytes of *S. exigua*. In addition to their immunity function as antimicrobial peptides, cecropins play a role in cuticle formation by regulating *PPO* expression during development [62]. PMH (paired mesoderm homeobox protein 2A-like) is a transcription factor of the homeobox gene family with crucial roles in development, tissue regeneration, and cellular differentiation [63]. Its expression was highly detected in plasmatocytes of *S. exigua* in our current assay. Its physiological function in hemocytes remains unknown but suggests a specific differentiation of the hemocytes during development. *PPO* expression is confined to oenocytoid in *S. exigua* [50]. Thus, the *PPO* gene expression was used as a FISH probe to determine the oenocytoid hemocytes. Just like the four oenocytoid clusters characterized by the *PPO* expression in *B. mori* [26], our four oenocytoid clusters also responded positively to the FISH probe. *Repat9* expression was used as a FISH probe to recognize spherulocyte. Repat (response to pathogen) is an immune-associated gene family encoded in *S. exigua* [64]. Repat proteins are relatively small (~15 kDa) and can interact with other Repat proteins [65]. For example, Repat1 is localized in the cytosol, but translocated into the nucleus in the presence of Repat8 [66]. Its functional domain possesses multi-bridge factor domain, which acts as a transcriptional co-activator and mediates gene expressions. In *S. exigua*, Repat33 mediates both cellular and humoral immune responses, presumably by activating immune-associate genes under eicosanoid signal [29]. Our current study indicates that *Repat9* expression is confined to spherulocyte, suggesting a specific immunological role for these hemocytes that was not well understood. The FISH probes developed would be valuable tools for various research purposes associated with hemocytes of *S. exigua*.

The molecular probes could discriminate the four hemocyte types without too much difference in their relative proportions compared to those obtained from the morphological classification. However, there was a marked reduction (>20%) in the relative proportion of the granulocytes detected in the FISH probe. Instead, the nHCs were detected in this study with almost 16%. This suggest that some of the granulocytes determined by the morphological characters are recognized by the molecular probe into nHCs. Here may be two hypotheses to explain the difference. One is that nHCs are derived from the granulocytes. Indeed, the lineage 2 of our trajectory analysis indicates the differentiation of 6 subtypes of granulocytes into nHC4. The other hypothesis is to regard the nHCs as the independent hemocyte type. This is supported by the clear distinct grouping in UMAP analysis of scRNA-Seq, in which nHCs are away from other hemocytes and independently differentiated upon the immune challenge. This observation is consistent with the possibility that some or all of the nHC populations represent transient differentiation or activation states of classical hemocyte types, particularly GRs, rather than entirely novel cell lineages. In our trajectory analysis, Lineage 2 (GR7 to nHC4) supports a scenario of trans-differentiation or intermediate differentiation states occurring within the hemolymph. Such state transitions have been documented in other insects, including *Drosophila*, where hemocytes exhibit plasticity and can adopt alternative fates under immune challenge [24]. Incorporating this interpretation provides a more balanced view of nHC identity, recognizing that these subpopulations may reflect functional diversification within existing lineages rather than the emergence of wholly new cell types. The functions displayed by the nHCs need to be clarified in subsequent study to understand the identity of the nHCs.

Hemocyte diversity revealed by the scRNA-Seq is ecologically significant in insects because it provides the immune flexibility to overcome various environmental stresses. The functional diversity of the 24 clusters in the immune responses was detected in their specific transcripts of the signaling components in Toll/IMD pathways, suggesting the unique and the interacting physiological roles among the different hemocyte types. This pathogen recognition signal led to the differentiated humoral and cellular immune responses such as AMP and PO production and apoptosis like other lepidopteran insects [67], in which our current study suggests the specific roles of the different hemocyte types. In addition to the immunological functions, the ability of different hemocyte types allows insects to adapt to various abiotic stressors in diverse ecological niches [68,69].

## 5. Conclusions

This study reports 24 hemocyte functional types of *S. exigua* based on the clustering analysis of the single-cell transcriptomes. Based on transcriptional variation and traditional morphological classification, larval hemocytes of *S. exigua* are re-categorized into five main types: GR, PL, OE, SP, and nHC. GRs (12–20 µm) are granular and phagocytic; PLs (10–15 µm) are spindle-shaped and adhesive; OEs (15–25 µm) are large, PPO-producing cells; SPs (15–25 µm) contain refractive spherules; and nHCs (10–18 µm) are smaller cells characterized by distinct marker genes. Among the 14,866 hemocytes for control naïve larvae analyzed via scRNA-Seq, their relative proportions are approximately 46% GR, 22% PL, 16% OE, 6% SP, and 16% nHC (with each of the five nHC subtypes comprising 3–4%). This sample represents an estimated 1.2–1.5% of the total hemocyte population, which is approximately 1.0–1.2 × 10^6^ cells in the L5 larvae. It also showed that these variable hemocytes are dynamic and undergo trans-differentiation after the maturation from the stem cells. Finally, this study provides several FISH probes to discriminate the different hemocyte types.

## Figures and Tables

**Figure 1 cells-14-01842-f001:**
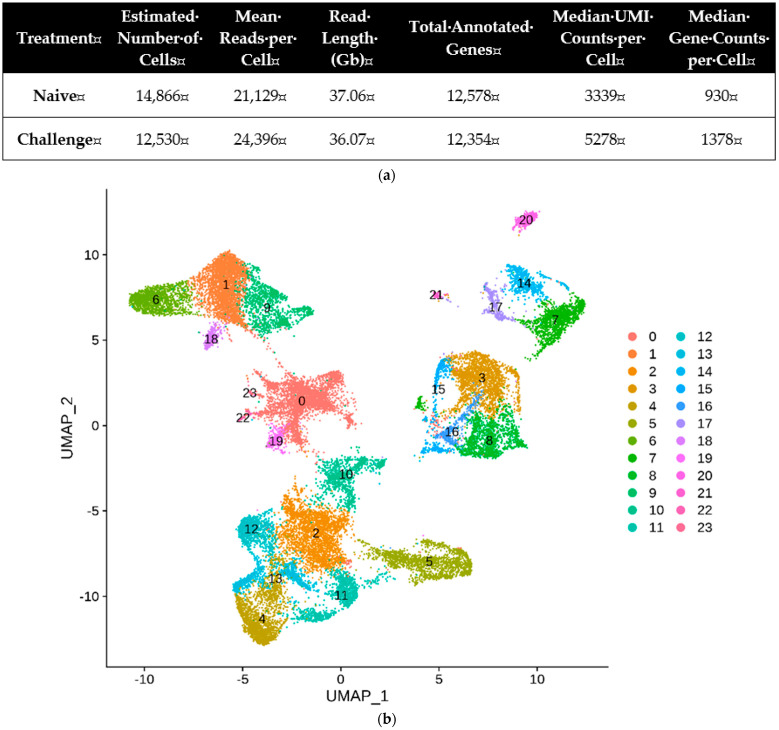
scRNA-Seq profiling of *S. exigua* hemocytes in naïve (‘Naïve’) and immune-challenged (‘Challenge’) larvae. (**a**) Summary of scRNA-Seq statistics. (**b**) UMAP indicating 24 distinct clusters (‘0–23’ in different colors) identified from pooled hemocytes of naïve and immune-challenged larvae. For the immune challenge, each L5 larva was injected with 1.4 × 10^5^ heat-killed *E. coli*, while PBS was injected for naïve controls. The numbers in the figure represent different clusters and each dot represents a single cell in each cluster, colored according to its assigned cluster.

**Figure 2 cells-14-01842-f002:**
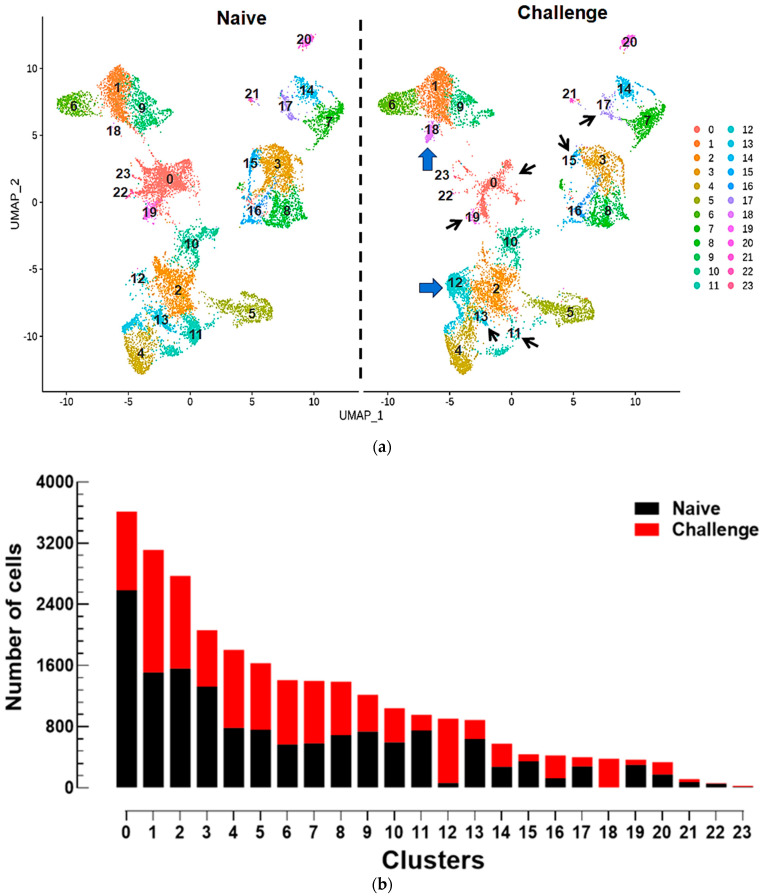
Alteration of scRNA-Seq clustering patterns of *S. exigua* hemocytes after immune challenge (‘Challenge’) compared to naïve larvae. (**a**) UMAP visualization of the altered hemocyte patterns in 24 clusters. Black thin arrows (0, 11, 13, 15, 17, and 19 clusters) indicate decreased cell numbers, while blue thick arrows (12 and 18 clusters) indicate increased cell numbers. (**b**) Bar plots showing the number of cells in each cluster for the two treatments. (**c**) Differential gene expression profiles after the immune challenge using unique genes (red line) and DEGs (black line) in each cluster. For the immune challenge, each L5 larva was injected with 1.4 × 10^5^ heat-killed *E. coli*, while PBS was injected for naïve controls.

**Figure 3 cells-14-01842-f003:**
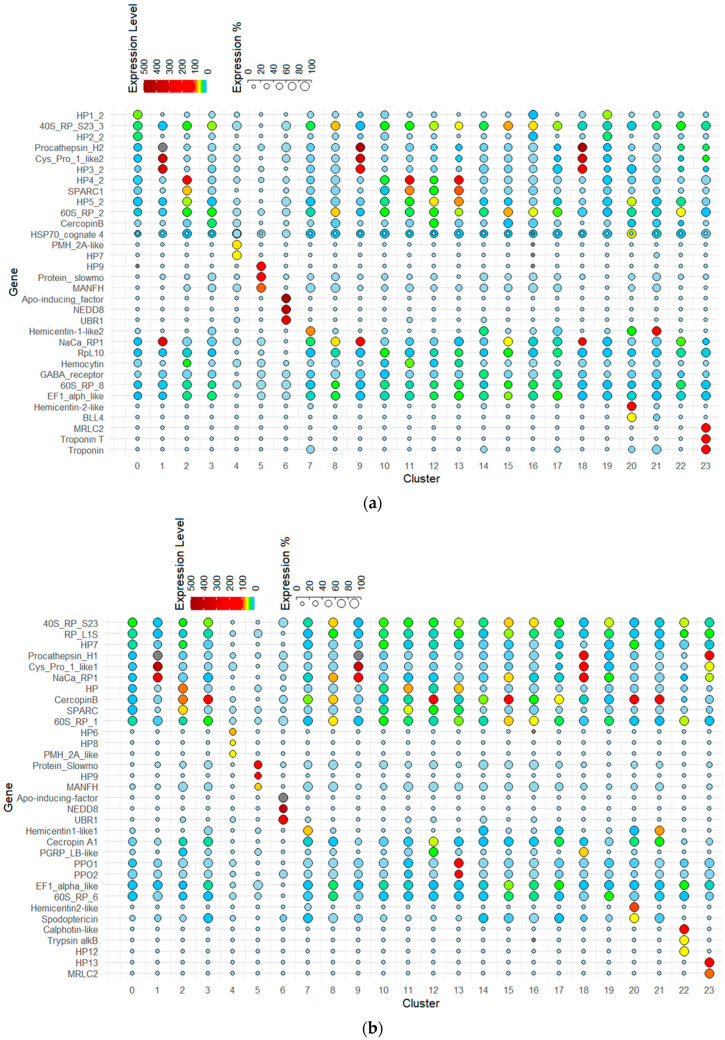
Dot plot represents the top three genes (*Y*-axis) enriched per cluster (*X*-axis) in hemocytes derived from naïve (**a**) and immune-challenged (**b**) larvae. Color gradient of the dot represents the expression level in FPKM, while the size represents percentage of cells expressing any gene per cluster. The complete dataset of cluster markers, including FPKM, adjusted *p*-values, and expression percentages, is provided in Appendix A for naïve and immune-challenged larvae, respectively.

**Figure 4 cells-14-01842-f004:**
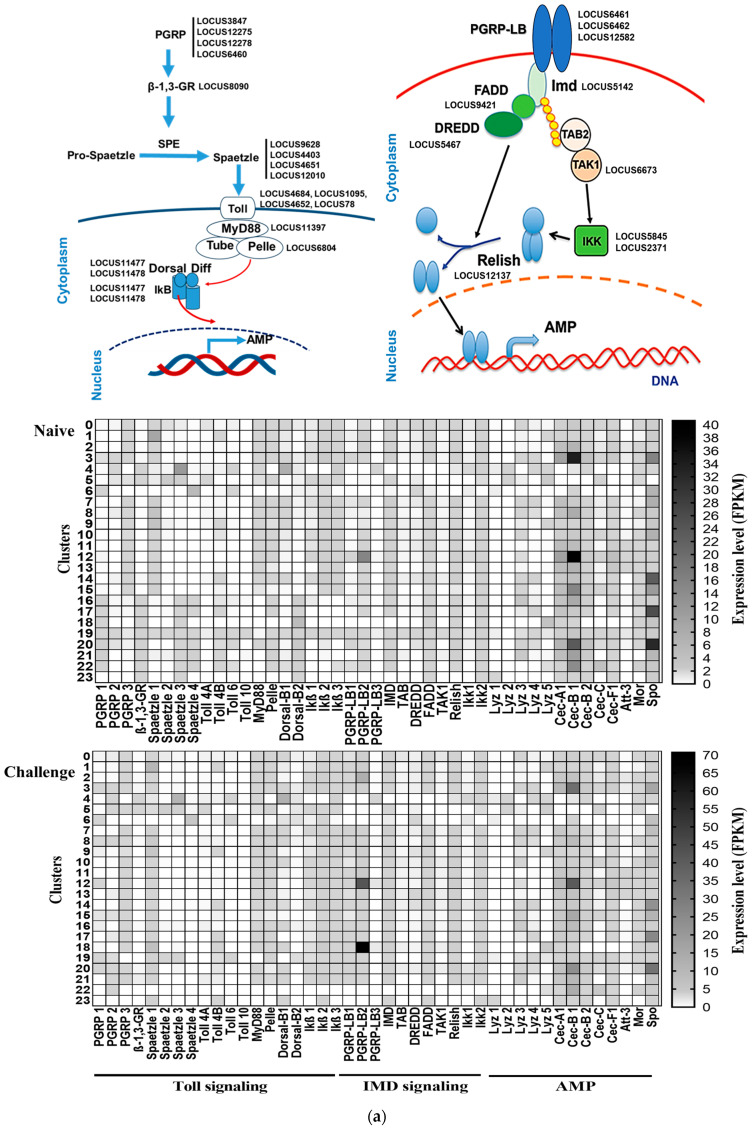
Specific expression profiles of the immune-associated genes in each of 24 scRNA-Seq clusters of naïve controls (‘Naïve’) and immune challenged (‘Challenge’) *S. exigua* larvae. For the immune challenge (‘Challenge’), each L5 larva was injected with 1.4 × 10^5^ heat-killed *E. coli*, while PBS was injected for Naïve. Heatmaps show expression patterns (based on FPKM) in naïve or immune challenge with different colors: white bar indicates no expression in either naïve or immune-challenged (“Challenge”) larvae. The expressed gene IDs are denoted in the corresponding components in the diagram. (**a**) Differential gene expression profiles of signal components associated with the Toll-IMD immune pathway and subsequent AMPs [1]: ‘Lyz’ for *lysozyme*, ‘Cec’ for *cecropin*, ‘Att’ for *attacin*, ‘Mor’ for *moricin*, and ‘Spo’ for *spodoptericin*. (**b**) Differential gene expression profiles of synthetic components associated with eicosanoid or epoxyoctadecamonoenoic acid (‘EpOME’) biosynthesis [29,35,36]: ‘PL’ for phospholipid, ‘LA’ for linoleic acid, ‘AA’ for arachidonic acid, ‘EET’ for epoxyeicosatrienoic acid, ‘DiHOME’ for dihydroxyoctadecanoic acid, ‘TX’ for thromboxane, ‘PG’ for prostaglandin, ‘HpETE’ for hydroperoxyeicosatetraenoic acid, ‘LT’ for leukotriene, ‘Elo/Des’ for *elongase*/*desaturase*, ‘Pxt’ for *peroxinectin*, ‘sEH’ for *soluble epoxide hydrolase*, ‘EPX’ for *epoxidase*, ‘EH’ for *epoxide hydrolase*, ‘PGDH/PGR’ for *prostaglandin dehydrogenase*/*reductase*, ‘LOX’ for *lipoxygenase*, ‘LTA_4_H’ for *LTA_4_ hydrolase*, and ‘Gpx’ for *glutathione peroxidase*. (**c**) Differential gene expression profiles of PGE_2_ signal components [37]: ‘R’ for *PGE_2_ receptor*, ‘Gαs-f’ for trimeric Gα subunits, ‘AC’ for *adenylate cyclase*, PLC for *phospholipase C*, ‘IP3’ for inositol-1,4,5-triphosphate, ‘IP3R’ for *IP3 receptor*, ‘ER’ for endoplasmic reticulum, ‘RyR’ for *ryanodine receptor*, ‘AQP’ for *aquaporin*, ‘PKA’ for *protein kinase A*, and ‘CICR’ for calcium-induced calcium release. (**d**) Differential gene expression profiles of prophenoloxidase (‘PPO’)-activating signal components [38]: ‘PO’ for *phenoloxidase*, ‘PAP’ for *PPO-activating protease*, ‘SPH’ for *serine protease homolog*, and ‘SPN’ for *serpin*. (**e**) Differential gene expression profiles of apoptosis signal components [13]: ‘MAPK’ for *mitogen-activated protein kinase*, ‘Cyt C’ for *cytochrome C oxidase*, ‘APAF’ for *apoptotic protease-activating factor*, ‘Cas’ for *caspase*, and ‘IAP’ for *inhibitor of apoptosis*.

**Figure 5 cells-14-01842-f005:**
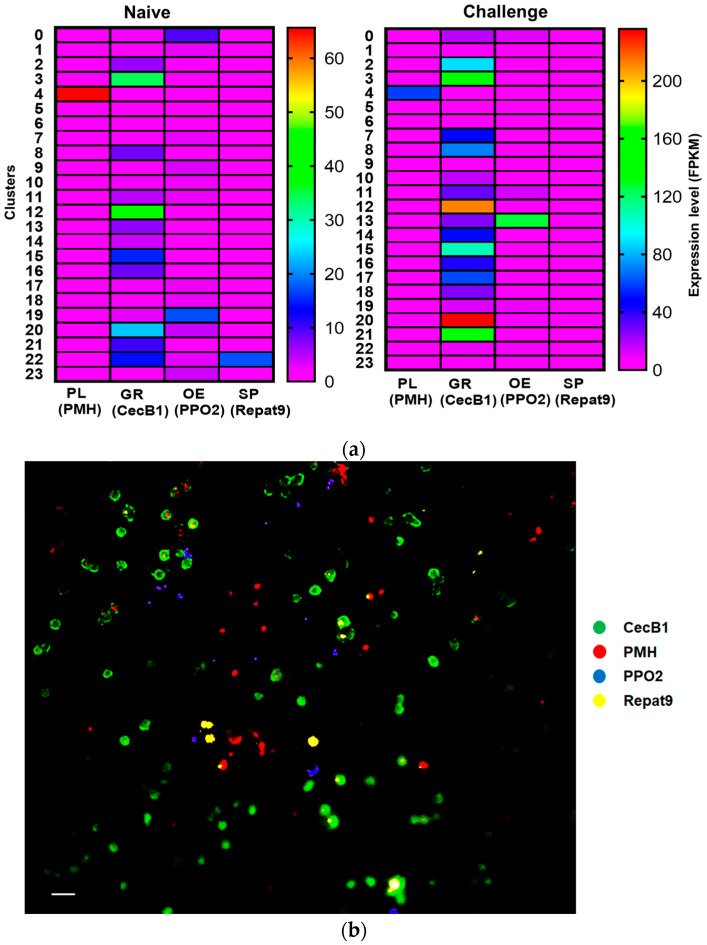
FISH probes specifically recognizing hemocyte types of *S. exigua* larvae. (**a**) Heatmap showing the specific expressions (average log2 fold change) of the marker genes across clusters between hemocytes collected from naïve (‘Naïve’) and immune-challenged (‘Challenge’) larvae. Marker genes include *cecropin B1* (‘CecB1’) for granulocyte (‘GR’), *paired mesoderm homeobox protein 2A-like* (‘PMH’) for plasmatocyte (‘PL’), *prophenoloxidase-2* (‘PPO2’) for oenocytoid (‘OE’), and *Repat9* for spherulocyte (‘SP’). (**b**) Specific recognition of the FISH probes to hemocytes collected from *S. exigua* larvae. (**c**) Efficiency of the hemocyte recognition by the FISH probes. Hemocytes are determined by the morphological characters [7,8] in DIC mode and recognized by FISH probes in FITC mode. Quantitative comparison between the frequencies of hemocyte types determined by morphological classification and FISH-positive cells detected by FISH probes. Different letters above bars indicate significant differences between groups (one-way ANOVA, *p* < 0.05). Finally, samples were mounted with a coverslip and observed under a fluorescence microscope (DM2500; Leica, Wetzlar, Germany) at ×200 magnification.

**Figure 6 cells-14-01842-f006:**
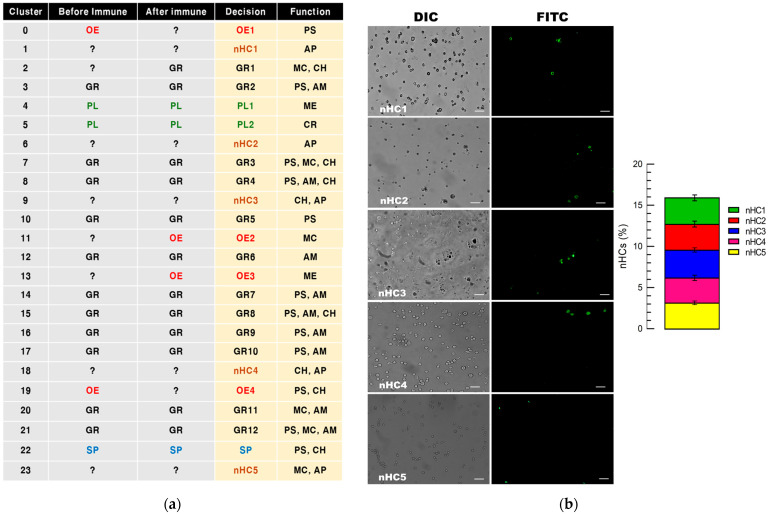
Comparison of the four morph-based hemocyte types and the 24 scRNA-Seq-based hemocyte clusters. (**a**) Annotation of 24 hemocyte clusters into classical and novel hemocyte types (denoted in different colors): 12 granulocyte types (‘GR1-GR12’), 2 plasmatocyte types (‘PL1-PL2’), 4 oenocytoid types (‘OE1-OE4’), a spherulocyte type (‘SP’), and five novel hemocyte types (‘nHC1-nHC5’). Functions of each cluster are assigned as follows: PS: protein synthesis; MC: modulating cell shape; AM: AMP production; CH: calcium homeostasis; AP: apoptosis; ME: melanization; CR: cell repairing. (**b**) Specific FISH markers for novel hemocyte (‘nHC’) types: *2-oxo-4-hydroxy-4-carboxy-5-ureidoimidazoline decarboxylase-like* (2-Oxo for ‘nHC1’), *Down syndrome* (DS for ‘nHC2’), *cysteine-proteinase 1-like* (CP1 for ‘nHC3’), *G-protein–coupled receptor No. 9* (GPCR No9 for ‘nHC4’), and *adhesion G protein–coupled receptor A3* (GPCR A3 for ‘nHC5’). Representative fluorescent images show probe-specific signals (green) in each subpopulation (probe sequences are provided in Appendix A). The bar plot (right) depicts the relative proportion of each nHC subpopulation among the total hemocytes. The bars indicate standard deviations of the nHC means. (**c**) Hemocyte differentiation in the hemocyte types after immune challenge in UMAP visualization. Each dot represents a single cell, colored by its assigned cluster, with numbers indicating cluster identities. Finally, samples were mounted with a coverslip and observed under a fluorescence microscope (DM2500; Leica, Wetzlar, Germany) at ×200 magnification.

**Figure 7 cells-14-01842-f007:**
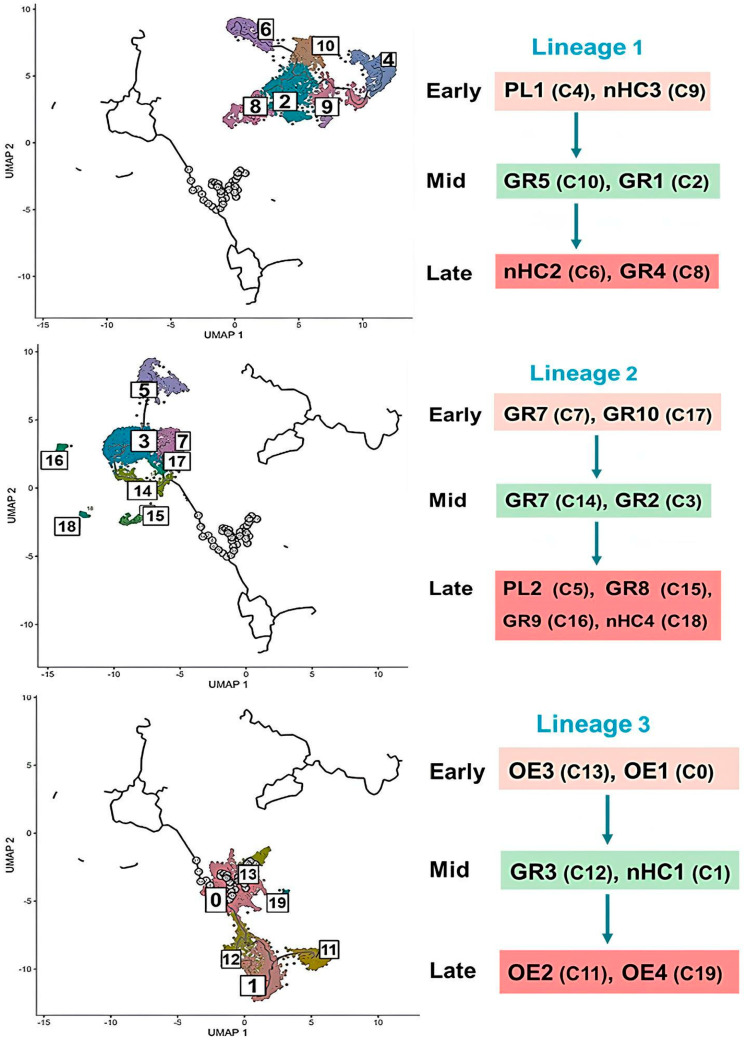
Pseudo-time trajectory analysis of 24 hemocyte clusters (0–23) of *S. exigua*. Pseudo-time trajectories were reconstructed from pooled hemocytes of naïve and immune-challenged larvae using Monocle 3 in R. Prohemocytes were designated as the root state, marking the earliest point of the differentiation pathway. Each cluster is color-coded as shown in the figure, with individual dots representing single cells positioned along the inferred developmental continuum. Hemocyte clusters were organized into three distinct lineages (Lineage 1–Lineage 3), with each lineage trajectory subdivided into three developmental stages: an early stage, enriched in progenitor-like clusters at the root; a mid-stage, consisting of transitional clusters positioned along branching trajectories; and a late stage, composed of terminally differentiated hemocyte subtypes located at branch termini.

## Data Availability

The original contributions presented in the study are included in the article/Appendix A; further inquiries can be directed to the corresponding author.

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
