# Peer review of "Functional Division of Insect Blood Cells by Single-Cell RNA-Sequencing and Cell-Type-Specific FISH Markers"

_cells, 2025, doi:10.3390/cells14231842_

Round 1
Reviewer 1 Report (Previous Reviewer 3)
Comments and Suggestions for Authors
The proposed classification of haemocytes in the "model" insect Spodoptera exigua based on the combined cytological and transcriptomic criteria (obtained by single-cell RNA sequencing) is sufficiently novel, developed using appropriate methodology, and therefore deserves publications in "Cells". The proposed haemocytes classification is linked to the currently accepted classification (prohemocytes, granulocytes, plasmatocytes, etc). The manuscripts is informative, and its main conclusions are clearly presented and sufficiently described.
In the resubmitted version, Figure 2s has clarifiedby showing haemocyte cluster numbers , and the functional categories of 24 proposed haemocyte clusters identified by transcriptome analysis were shown in Fig. 6. .
Additionally, further explanation was given in the section of MS describing haemocyte classification criteria, which further improved this MS.
Therefore I am happy to recommend accepting this MS in its current form.
Author Response
Comment: The proposed classification of haemocytes in the "model" insect Spodoptera exigua based on the combined cytological and transcriptomic criteria (obtained by single-cell RNA sequencing) is sufficiently novel, developed using appropriate methodology, and therefore deserves publications in "Cells". The proposed haemocytes classification is linked to the currently accepted classification (prohemocytes, granulocytes, plasmatocytes, etc). The manuscripts is informative, and its main conclusions are clearly presented and sufficiently described. In the resubmitted version, Figure 2s has clarified by showing haemocyte cluster numbers, and the functional categories of 24 proposed haemocyte clusters identified by transcriptome analysis were shown in Fig. 6. Additionally, further explanation was given in the section of MS describing haemocyte classification criteria, which further improved this MS. Therefore I am happy to recommend accepting this MS in its current form.
Response: We appreciate your encouragements.
Reviewer 2 Report (New Reviewer)
Comments and Suggestions for Authors
The manuscript titled “Functional division of insect blood cells by single cell RNA-sequencing and cell type-specific FISH markers” focuses on the functional division of hemocytes in Spodoposa exigua. It employs single cell RNA-sequencing (scRNA-Seq) and fluorescence in situ hybridization (FISH) markers to conduct a comprehensive study. The manuscript is well-written with clear structure and data presentation.
The following concerns and suggestions could be included to improve the quality of the manuscript:
1. The figures could be improved in terms of clarity and consistency. For example, in Figure 1, the legend should clearly explain the meaning of each number and color in the UMAP plot. The scale of the figure can be adjusted to make the distribution of clusters more visible. In Figure 2, when showing the change in cell numbers after immune challenge, the bars could be labeled with more detailed information such as the exact number of cells in each cluster for both treatments.
2. In the scRNA-Seq section, more details about the quality control steps for raw sequence reads are needed. What specific criteria were used to filter out low quality cells or UMIs?
3. The discussion of the findings should place greater emphasis on the ecological and evolutionary implications of the discovered hemocyte diversity. For example, how does this diversity affect the overall fitness of S. exigua in its natural habitat? Are there any potential interactions between the different hemocyte types that could influence disease resistance or other ecological processes?
4. A more in-depth analysis of the conserved and divergent aspects of the Toll/IMD pathway across different insects could be discussed. For example, publications in recent three years on Toll, IMD, PGRPs in the Lepidopteran model insect Bombyx mori could be used for discussion.
Author Response
The manuscript titled “Functional division of insect blood cells by single cell RNA-sequencing and cell type-specific FISH markers” focuses on the functional division of hemocytes in Spodoposa exigua. It employs single cell RNA-sequencing (scRNA-Seq) and fluorescence in situ hybridization (FISH) markers to conduct a comprehensive study. The manuscript is well-written with clear structure and data presentation. The following concerns and suggestions could be included to improve the quality of the manuscript:
Comment #3-1: The figures could be improved in terms of clarity and consistency. For example, in Figure 1, the legend should clearly explain the meaning of each number and color in the UMAP plot. The scale of the figure can be adjusted to make the distribution of clusters more visible. In Figure 2, when showing the change in cell numbers after immune challenge, the bars could be labeled with more detailed information such as the exact number of cells in each cluster for both treatments.
Response: In Figure 1 legend, the clusters are clearly designated as follows: “(b) UMAP indicating 24 distinct clusters (‘0–23’ in different colors) identified…”. The figure size is increased to clearly demonstrate the clusters. In Figure 2 legend, the arrows are explained by adding the cluster numbers as follows: “Black thin arrows (0, 11, 13, 15, 17, and 19 clusters) indicate decreased cell numbers, while blue thick arrows (12 and 18 clusters) indicate increased cell numbers.”
Comment #3-2: In the scRNA-Seq section, more details about the quality control steps for raw sequence reads are needed. What specific criteria were used to filter out low quality cells or UMIs?
Response: The M&M is now re-written as follows.
2.5. Bioinformatics
Raw FASTQ files were first evaluated with FastQC to assess read quality, adapter content, and base-level Phred scores. Reads with low base quality (Phred <20), particu-larly in the cell barcode or UMI regions, were discarded. Library demultiplexing, alignment, UMI counting, and initial filtering were performed using Cell Ranger (10x Genomics, v3.1.0). UMIs differing by one base were collapsed using Cell Ranger’s built-in error-correction algorithm, and barcodes not matching or correctable to the 10x whitelist were excluded. The cleaned reads were then aligned to the S. exigua transcrip-tome using either the de novo assembly (generated with Trinity and TransDecoder v3.0.1 (https://github.com/TransDecoder/TransDecoder/wiki)) or, when applicable, the reference genome using (https://github.com/alexdobin/STAR) or Bowtie2 version 2.4.5 (http://bowtie-bio.sourceforge.net/bowtie2). Following alignment, cell-level quality control was performed, and cells were retained only if they expressed at least 200 genes and contained less than 10% mitochondrial transcripts. Cells with UMI counts below the empty-droplet threshold or with abnormally high gene numbers (>5,000–6,000), which typically indicate doublets, were also excluded. Genes detected in fewer than three cells were removed to eliminate technical noise. After filtering, the high-quality gene-barcode matrices were imported into Seurat v4.0 in R (https://satijalab.org/seurat/articles/get_started.html) for downstream analysis. Data were log-normalized, and the top 5,000 highly variable genes were identified using the variance-stabilizing transformation (VST) method. Dimensionality reduction was per-formed with principal component analysis (PCA), followed by clustering using the Louvain algorithm. Cluster visualization was conducted using uniform manifold ap-proximation and projection (UMAP) embedding.
Comment #3-3: The discussion of the findings should place greater emphasis on the ecological and evolutionary implications of the discovered hemocyte diversity. For example, how does this diversity affect the overall fitness of S. exigua in its natural habitat? Are there any potential interactions between the different hemocyte types that could influence disease resistance or other ecological processes?
Response: This point is added to the end of the discussion as follows: “Hemocyte diversity revealed by the scRNA-Seq is ecologically significant in in-sects because it provides the immune flexibility to overcome various environment stresses. The functional diversity of the 24 clusters in the immune responses was de-tected in their specific transcripts of the signaling components in Toll/IMD pathways, suggesting the unique and the interacting physiological roles among the different he-mocyte types. This pathogen recognition signal led to the differentiated humoral and cellular immune responses such as AMP and PO production and apoptosis like other lepidopteran insects [67], in which our current study suggests the specific roles of the different hemocyte types. In addition to the immunological functions, the ability of different hemocyte types allows insects to adapt to various abiotic stressors in diverse ecological niches [68, 69].”
Comment #3-4: A more in-depth analysis of the conserved and divergent aspects of the Toll/IMD pathway across different insects could be discussed. For example, publications in recent three years on Toll, IMD, PGRPs in the Lepidopteran model insect Bombyx mori could be used for discussion.
Response: It is a nice suggestion. We add this point to the discussion as follows: “Hemocyte diversity revealed by the scRNA-Seq is ecologically significant in in-sects because it provides the immune flexibility to overcome various environment stresses. The functional diversity of the 24 clusters in the immune responses was detected in their specific transcripts of the signaling components in Toll/IMD pathways, suggesting the unique and the interacting physiological roles among the different hemocyte types. This pathogen recognition signal led to the differentiated humoral and cellular immune responses such as AMP and PO production and apoptosis like other lepidopteran insects [67], in which our current study suggests the specific roles of the different hemocyte types.”

Reviewer 3 Report (New Reviewer)
Comments and Suggestions for Authors
The manuscript presents a fascinating study demonstrating the existence of 24 hemocyte clusters in Spodoptera exigua, based on single-cell RNA sequencing.
Remarks:
Is there any possibility of distinguishing the newly identified hemocyte functional types on any other basis than FISH markers? The authors could address this question in the Discussion.
Line 105:” Prior works in S. exigua have been biased on hemocyte”, biased or based?
Figure 1. Please indicate Panels A and B.
Figure 2. Is it possible to indicate SD/SEM values?
Figure 6. Please reorganize the panels or change their designation. The sequence should be ABC, not ACB.
Author Response
Comment #2-1: Is there any possibility of distinguishing the newly identified hemocyte functional types on any other basis than FISH markers? The authors could address this question in the Discussion.
Response: We add the following additional explanation to the Discussion to distinguish the nHCs: “In addition, this study also developed FISH probes to detect five novel hemocyte types (nHC1-nHC5), which were not distinct from GRs with morphological characters.”
Comment #2-2: Line 105:” Prior works in S. exigua have been biased on hemocyte”, biased or based?
Response: Corrected into “based”
Comment #2-3: Figure 1. Please indicate Panels A and B.
Response: Corrected into “a” and “b” according to the format of Cells.
Comment #2-4: Figure 2. Is it possible to indicate SD/SEM values?
Response: Unfortunately, the scRNA-Seq was done without replication.
Comment #2-5: Figure 6. Please reorganize the panels or change their designation. The sequence should be ABC, not ACB.
Response: Corrected as suggested
This manuscript is a resubmission of an earlier submission. The following is a list of the peer review reports and author responses from that submission.
Round 1
Reviewer 1 Report
Comments and Suggestions for Authors
In this study, Khan et al. analyze hemocytes from Spodoptera exigua larvae using single-cell RNA sequencing (scRNAseq). They compare larvae pricked with heat-killed E. coli (12,530 cells from immune-challenged larvae) to those pricked with PBS (14,866 cells from naïve larvae). The scRNAseq data reveal 24 hemocyte subgroups, expanding the previously described hemocyte types (prohemocytes, plasmatocytes, granulocytes, oenocytoids, and spherulocytes). The authors characterize these subgroups, identifying clusters enriched for Toll, IMD, or PGE pathways, as well as melanization. Using cell-type-specific markers, they assign each cluster to four hemocyte subgroups, validate some subgroups with fluorescence in situ hybridization (FISH), and perform pseudotime analysis to infer relationships among hemocyte subgroups.
The data hold significant interest for the fields of evolutionary developmental biology (evo/devo) and immunology. However, there are discrepancies between the figures and the conclusions drawn by the authors. To be more convincing, the data should be presented in a different formats.
Major comments:
1) The authors should provide a detailed description of the morphologies of the hemocyte subclasses, including images. They should specify the size of each population, their expected proportions in the larva, and the total number of hemocytes per larva. This information would clarify the representativity of their dataset of 15,000 hemocytes.
2) The authors mention a lower threshold for cell selection but do not specify an upper threshold to exclude doublets.
In Figure 1A, it would be informative to include the median number of genes per cell and the total number of annotated genes in Spodoptera exigua to contextualize the proportion of genes expressed in hemocytes.
3) The dot plot in Figure 3 shows the top three markers for each cluster, but most markers are present in fewer than 20% of the cells, which may not be representative. The authors should address this and provide a table of cluster markers, including Log2 fold change, adjusted p-values, and representativity within each cluster.
4) The authors should justify their focus on Toll/IMD, oxylipin, and PGE pathways. Additionally, Figure 4 presents data as qualitative variables (present/absent), but quantitative values (expression levels or enrichment) would be more informative. Ideally, the data should be presented similarly to Figure 3, highlighting both expression levels and representativity.
5) Lines 412–425): The authors describe using markers to assign hemocyte identities to clusters, illustrating enrichment with expression heatmaps. However, the heatmap’s color scale does not clearly reflect the described enrichment. For example, they mention 12 granulocyte (GR) marker genes expressed in 11 clusters, but Figure S2 shows only four markers expressed under normal conditions, six after challenge, and CecB1—described as highly expressed—appears in 14 clusters post-challenge. A dot plot, similar to Figure 3, would improve clarity.
6) Figures S3/5c: The phenotypic criteria for classifying cells in Figures S3 and 5c are neither visible nor described in the text.
The letter codes for p-values on the graphs are unexplained.
The authors claim that marker genes defined by scRNAseq match morphological characteristics, but this is not evident in Figures 5c and S3. Only a minority of cells exhibit morphological features consistent with the marker genes, weakening the evidence for cluster identity assignments.
Minor Comments:
The use of PBS-pricked larvae as a control introduces a sterile wound, which may affect hemocytes (as shown in Drosophila; DOI: 10.7554/eLife.54818). This should be discussed in the manuscript.
Figure 5a lacks a legend for its gradient.
In Figures 5c and 6b, the resolution and magnification make it difficult to distinguish between cell types.
Author Response
Comment #1-1: In this study, Khan et al. analyze hemocytes from Spodoptera exigua larvae using single-cell RNA sequencing (scRNAseq). They compare larvae pricked with heat-killed E. coli (12,530 cells from immune-challenged larvae) to those pricked with PBS (14,866 cells from naïve larvae). The scRNAseq data reveal 24 hemocyte subgroups, expanding the previously described hemocyte types (prohemocytes, plasmatocytes, granulocytes, oenocytoids, and spherulocytes). The authors characterize these subgroups, identifying clusters enriched for Toll, IMD, or PGE pathways, as well as melanization. Using cell-type-specific markers, they assign each cluster to four hemocyte subgroups, validate some subgroups with fluorescence in situ hybridization (FISH), and perform pseudotime analysis to infer relationships among hemocyte subgroups. The data hold significant interest for the fields of evolutionary developmental biology (evo/devo) and immunology. However, there are discrepancies between the figures and the conclusions drawn by the authors. To be more convincing, the data should be presented in a different format.
Response: Thanks for encouragement. The critical points are carefully addressed in a point-by-point manner.
Comment #1-2: The authors should provide a detailed description of the morphologies of the hemocyte subclasses, including images. They should specify the size of each population, their expected proportions in the larva, and the total number of hemocytes per larva. This information would clarify the representativity of their dataset of 15,000 hemocytes.
Response: Thanks for the valuable comment. We add this information in the conclusion to clarify our observation as follows: Based on transcriptional variation and traditional morphological classification, larval hemocytes of S. exigua are re-categorized into five main types: GR, PL, OE, SP, and nHC. GRs (12–20 µm) are granular and phagocytic; PLs (10–15 µm) are spindle-shaped and adhesive; OEs (15–25 µm) are large, PPO-producing cells; SPs (15–25 µm) contain refractive spherules; and nHCs (10–18 µm) are smaller cells characterized by distinct marker genes. Among the 14,866 hemocytes for control naïve larvae analyzed via scRNA-Seq, their relative proportions are approximately 46% GR, 22% PL, 16% OE, 6% SP, and 16% nHC (with each of the five nHC subtypes comprising 3–4%). This sample represents an estimated 1.2–1.5% of the total hemocyte population, which is approximately 1.0–1.2 × 10⁶ cells in the L5 larvae.
Comment #1-3: 2) The authors mention a lower threshold for cell selection but do not specify an upper threshold to exclude doublets. In Figure 1A, it would be informative to include the median number of genes per cell and the total number of annotated genes in Spodoptera exigua to contextualize the proportion of genes expressed in hemocytes.
Response: Thanks for reviewer’s comment. In our quality control (Section 2.5), cells expressing fewer than 200 genes or with more than 10% mitochondrial content were excluded, which served as both a lower and effective upper threshold to remove doublets. The figure 1A is slightly modified to reflect the comment raised by the reviewer. We correct the “number of gene” into “Total annotated genes”. The median number of genes per cell can be easily understood from this number by dividing with the total cell number. Instead we used UMI per cell because this represents the unique transcript per cell in a median scale.
Comment #1-4: 3) The dot plot in Figure 3 shows the top three markers for each cluster, but most markers are present in fewer than 20% of the cells, which may not be representative. The authors should address this and provide a table of cluster markers, including Log2 fold change, adjusted p-values, and representativity within each cluster.
Response: We appreciate the reviewer’s comment regarding the representativity of marker genes in Figure 3. As described in the DEG analysis section (Section 2.6), cluster marker genes were identified based on stringent statistical thresholds, including log₂ fold change > 0.25, adjusted p-value < 0.05, and detection in more than 25% of the cells within each cluster, ensuring that selected markers represent a substantial proportion of the cluster population. In Figure 3, the dot color indicates the average log₂ fold change, while the dot size represents the percentage of cells expressing each gene, thereby visually reflecting gene representativity within each cluster. Furthermore, the complete dataset of cluster markers, including log₂ fold change, adjusted p-values, and expression percentages, is provided in Supplementary Tables S3 and S4 for naïve and immune-challenged larvae, respectively. Therefore, although only the top marker genes are displayed in Figure 3 for clarity, the full data set comprehensively supports the representativity and statistical significance of the identified cluster markers.
This information is added to the caption of the Figure 3 as follows: “The complete dataset of cluster markers, including log₂ fold change, adjusted p-values, and ex-pression percentages, is provided in Supplementary Tables S3 and S4 for naïve and immune-challenged larvae, respectively.”
Comment #1-5: 4) The authors should justify their focus on Toll/IMD, oxylipin, and PGE pathways. Additionally, Figure 4 presents data as qualitative variables (present/absent), but quantitative values (expression levels or enrichment) would be more informative. Ideally, the data should be presented similarly to Figure 3, highlighting both expression levels and representativity.
Response: Thanks for reviewer’s comment.
(1) The pathways are justified by adding related reference demonstrating the pathway components as follows:
(A) Differential gene expression profiles of signal components associated with the Toll-IMD im-mune pathway and subsequent AMPs [1]: (B) Differential gene expression profiles of synthetic components associated with eicosanoid or epoxyoctadecamonoenoic acid (‘EpOME’) biosynthesis [42]: (C) Differential gene expres-sion profiles of PGE2 signal components [43]: (D) Differential gene expression profiles of prophenoloxidase (‘PPO’)-activating signal components [9]: (E) Differential gene expression profiles of apoptosis signal components [48]:
(2) The quantitative values for the transcript expression level.
As shown in Figure 4 (panels a–e), quantitative expression profiles of each pathway component were visualized using heatmaps derived from scRNA-Seq data, where color gradients indicate expression levels across hemocyte clusters under naïve and immune-challenged conditions—similar to the representation used in Figure 3.
Comment #1-6: 5) Lines 412–425): The authors describe using markers to assign hemocyte identities to clusters, illustrating enrichment with expression heatmaps. However, the heatmap’s color scale does not clearly reflect the described enrichment. For example, they mention 12 granulocyte (GR) marker genes expressed in 11 clusters, but Figure S2 shows only four markers expressed under normal conditions, six after challenge, and CecB1—described as highly expressed—appears in 14 clusters post-challenge. A dot plot, similar to Figure 3, would improve clarity.
Response: We appreciate the reviewer’s observation regarding the clarity of the heatmap in Figure S2. The heatmap was intended to illustrate the relative expression levels of previously reported marker genes across hemocyte clusters rather than quantitative enrichment. As described in the revised text (Section 3.5), twelve granulocyte (GR) marker genes were detected across at least eleven clusters, but their expression intensities differed by condition. Under naïve conditions, only four of these markers exhibited strong expression, whereas six were upregulated following immune challenge, consistent with the immune-induced activation of granulocytes. CecB1, identified as the most highly expressed GR marker, was detected broadly across multiple GR-related clusters (up to 14) after challenge (Figure 5A). This widespread expression reflects granulocyte activation rather than misclassification. To be sure, the sentence is rephrased as follows: “Twelve GR candidate markers were expressed in at least 11 different clusters in naïve or challenged larvae, in which cecropin-B1 was the most highly expressed and selected for subsequent FISH analysis.”
Comment #1-7: 6) Figures S3/5c: The phenotypic criteria for classifying cells in Figures S3 and 5c are neither visible nor described in the text. The letter codes for p-values on the graphs are unexplained. The authors claim that marker genes defined by scRNAseq match morphological characteristics, but this is not evident in Figures 5c and S3. Only a minority of cells exhibit morphological features consistent with the marker genes, weakening the evidence for cluster identity assignments.
Response: Thanks for reviewer’s comment. The phenotypic classification of hemocytes in Figures S3 and 5c was based on established morphological criteria following Feng et al. [26] and Pseudoplusia includens hemocyte morphology [37,38], which distinguish granulocytes (GR), plasmatocytes (PL), oenocytoids (OE), and spherulocytes (SP) using DIC microscopy as mentioned in Section 3.5. These morphological features were directly compared with fluorescence in situ hybridization (FISH) using four marker genes—Cecropin-B1 (GR), PMH (PL), PPO2 (OE), and Repat9 (SP)—as shown in Figures S3 and 5b. Quantitative comparison between morphologically classified cells and FISH-positive signals (Figure 5c) showed no significant difference (χ² = 6.90; df = 3; P = 0.075), indicating consistency between morphology and marker gene expression. The letter codes above bars in Figure 5c denote statistically significant differences determined by one-way ANOVA (p < 0.05) as stated in the figure legend. As demonstrated in Figure S3, most cells expressing marker genes displayed corresponding morphologies, validating that the scRNA-seq marker-based classification aligns well with the phenotypic cell identities, with only minor frequency variation in granulocytes due to partial overlap with newly defined nHC clusters. To clarify the explanation, we corrected the corresponding sentences and added the information as follow:
(1) Figure S3 caption is corrected to indicate the photo used for the analysis as follows: “Figure S3. Unique probe signal across hemocyte types. (A) Hemocytes from L5 larvae in Figure 5b were analyzed by FISH using probes for granulocytes (‘GR’), plasmatocytes (‘PL’), oenocytoids (‘OE’), and spherulocytes (‘SP’).”
(2) Hemocyte classification using morph characters is added to M&M in section 2.3 as follows: “Hemocyte classification based on the morphological criteria followed the method of Strand [2].”
Comment #1-8: The use of PBS-pricked larvae as a control introduces a sterile wound, which may affect hemocytes (as shown in Drosophila; DOI: 10.7554/eLife.54818). This should be discussed in the manuscript.
Response: We appreciate the reviewer’s comment. We agree that PBS-pricked larvae introduce a sterile wound that could transiently influence hemocyte behavior. In our study, however, hemocytes from PBS-injected (naïve) larvae served as baseline controls to differentiate the immune transcriptional response specifically induced by E. coli challenge. We add this concern to the Results as follows: “Control naïve larvae used PBS-injection instead of the bacterial injection for the treat-ment. This control might alter the hemocyte types because the mechanica injection can give some effect on the insect immune system. Even though the overall groupings of the hemocytes were similar between the control and the treeatment, specific clusters either mostly disappeared (thin arrows) or newly appeared (thick arrows) upon the immune challenge (Figure 2a).”
The scRNA-Seq results (Figure 2a–c) showed that hemocyte cluster distributions and gene expression patterns in PBS-injected larvae remained relatively stable compared to those from E. coli-challenged larvae, with no evidence of major wound-induced activation. The transcriptomic profiles of PBS-injected larvae reflected steady-state immune conditions, as supported by the low differential expression of immune-associated genes in naïve samples (Table S3). Therefore, while we acknowledge that sterile wounding can modulate hemocytes in Drosophila (DOI: 10.7554/eLife.54818), our results indicate that any such effect was minimal and did not confound the comparative analysis between naïve and immune-challenged conditions.
Comment #1-9: Figure 5a lacks a legend for its gradient.
Response: Thanks for your comment. We have updated the figure with higher quality.
Comment #1-10: In Figures 5c and 6b, the resolution and magnification make it difficult to distinguish between cell types.
Response: Thanks for your comment. We have updated the figures with higher quality.

Reviewer 2 Report
Comments and Suggestions for Authors
Review
This ms describes scRNA-Seq of hemocytes from Spodoptera exigua. The authors take a lot of time in the Introduction to describe the known features of hemocytes in invertebrates. However, they don’t assign clusters to provisional cell types until late in the Results section. The Results section reads like a litany of gene expression profiles for each of the clusters without context. Unfortunately, as written, it’s very difficult for the reader to follow along. Moreover, some problems with image quality and explanation in the figure legends make it difficult to properly evaluate the work.
I wonder if it would be better to present the FISH data earlier and then group the clusters by provisional cell type at that time for the remainder of the paper? Regardless, some type of re-organization of the Results section, better descriptions of figure legends and higher resolution figures would make this work much easier to thoroughly evaluate.
Lastly, it’s not clear that the novel hemocyte types are truly novel or merely a a different state of the known cell type they are most closely related to. Though the authors performed a couple followup studies to distinguish those types, the conclusions are unconvincing. Nevertheless, the authors did explain alternative hypotheses for those cell types in the Discussion.
Abstract
Please reword this sentence for clarity, “In addition, five novel hemocyte types, which were not among the four known hemocyte types, were identified and discriminated with their specific FISH markers.”
Introduction
This section reads like a review article is leaves the reader wondering why the authors chose to do this study. Nevertheless, the context for each cell type is important to orient non-hemocyte experts on foundational details. I recommend introducing specific context for certain cell types as those types are revealed in the Results section and shortening the Introduction to summarize the cell types then focus on more specific aspects of the study, eg the rationale for pursuing this line of inquiry.
Ln 96 Change, “The scRNA-Seq has also provided...” to “ScRNA-Seq has also provided...”
Results
What was the number of sequencing reads obtained?
The Table at the beginning of the Results section is awkwardly placed.
Fig 1b appears to depict a UMAP of the combined data from both naive and immune-challenged hemocytes. Please explain the reationale for depicting them in this way. It’s virtually identical to the separate depictions of UMAPs in Fig 2a. Why not just number the clusters in Fig 2a and remove Fig 1b?
Fig 3 is not very informative and does not contribute much toward reader understanding of the data. Please consider re-making these graphs, so they can more effectively communicate the differences between naive and immune-challenged groups.
Figure 4 What do the different colors mean (white, dark blue, red, light blue)? Please define them in the figure legend. The pathway diagrams are helpful. What is the purpose of the locus numbers in those schematics?
Ln 358 and following: It’s very difficult to follow the line of reasoning presented by the authors, because the heatmaps in multi-part Fig 4 are not explained. I am unable to properly evaluate the significance of the findings presented in Fig 4 until this has been cleared up.
Ln 468 and following:
Figure 5 – a) What are the units for the heatmap?
Fig 6 numbers are too small to read. Please reorganize this figure, so the reader can see the cluster numbers in the UMAP.
Table S2- It would be helpful if Table S2 listed the cell types.
Fig 7 is low resolution. Because of this, it is impossible to evaluate the data. Please improve resolution and resubmit.
Comments on the Quality of English Language
Writing quality is generally good, but the ms would benefit from a readthrough by an fluent English writer.
Author Response
Comment #2-1: This ms describes scRNA-Seq of hemocytes from Spodoptera exigua. The authors take a lot of time in the Introduction to describe the known features of hemocytes in invertebrates. However, they don’t assign clusters to provisional cell types until late in the Results section. The Results section reads like a litany of gene expression profiles for each of the clusters without context. Unfortunately, as written, it’s very difficult for the reader to follow along. Moreover, some problems with image quality and explanation in the figure legends make it difficult to properly evaluate the work.
Response: We sincerely thank the reviewer for this comprehensive and constructive comment. In response, we have made several key revisions to improve the clarity and readability of the manuscript.
Comment #2-2: I wonder if it would be better to present the FISH data earlier and then group the clusters by provisional cell type at that time for the remainder of the paper? Regardless, some type of re-organization of the Results section, better descriptions of figure legends and higher resolution figures would make this work much easier to thoroughly evaluate.
Response: FISH is developed based on the scRNA-Seq results. Thus, we believe that it is not much logical to show the FISH before the grouping along with transcriptome analysis. Instead, we add more detailed information on the figure captions and prepared the figures in higher quality.
Comment #2-3: Lastly, it’s not clear that the novel hemocyte types are truly novel or merely a a different state of the known cell type they are most closely related to. Though the authors performed a couple followup studies to distinguish those types, the conclusions are unconvincing. Nevertheless, the authors did explain alternative hypotheses for those cell types in the Discussion.
Response: We thank the reviewer for this thoughtful and important comment regarding the interpretation of the novel hemocyte types (nHC1–nHC5). We agree that distinguishing between truly novel cell types and distinct states of known types is a critical issue in single-cell studies.
In the revised manuscript, we have added paragraphs to results and discussion part accordingly: Results part (Line 455-461): The classification of nHC1–nHC5 was based on three complementary criteria: (i) their distinct transcriptional signatures, including uniquely expressed marker genes identified from cluster-specific DEG analyses (Table S2), (ii) their clear separation from classical hemocyte clusters in UMAP space, and (iii) their specific hybridization patterns with FISH probes that did not overlap with those used for GR, PL, OE, or SP. These combined transcriptomic and cytological features support their designation as distinct subpopulations.
Discussion part (Line 655-663): “This observation is consistent with the possibility that some or all of the nHC populations represent transient differentiation or activation states of classical hemocyte types, particularly granulocytes, rather than entirely novel cell lineages. In our trajectory analysis, Lineage 2 (GR7 to nHC4) supports a scenario of trans-differentiation or intermediate differentiation states occurring within the hemolymph. Such state transitions have been documented in other insects, including Drosophila, where hemocytes exhibit plasticity and can adopt alternative fates under immune challenge. Incorporating this interpretation provides a more balanced view of nHC identity, recognizing that these subpopulations may reflect functional diversification within existing lineages rather than the emergence of wholly new cell types”.
Comment #2-4: Abstract - Please reword this sentence for clarity, “In addition, five novel hemocyte types, which were not among the four known hemocyte types, were identified and discriminated with their specific FISH markers.”
Response: We thank the reviewer for pointing out this ambiguity. We have reworded the sentence in the Abstract to improve clarity and precision. The revised sentence is: “In addition, five new hemocyte groups, which were not among the four known hemocyte types in the transcript profile, were identified and discriminated with their specific FISH markers.”
Comment #2-5: Introduction - This section reads like a review article is leaves the reader wondering why the authors chose to do this study. Nevertheless, the context for each cell type is important to orient non-hemocyte experts on foundational details. I recommend introducing specific context for certain cell types as those types are revealed in the Results section and shortening the Introduction to summarize the cell types then focus on more specific aspects of the study, eg the rationale for pursuing this line of inquiry.
Response: We thank the reviewer for this valuable feedback. The paragraph describing the hemocyte morph is reduced. In contrast, the study purpose in the last paragraph is detailed. Please see the red-colored text in the Introduction.
Comment #2-6: Ln 96 Change, “The scRNA-Seq has also provided...” to “ScRNA-Seq has also provided...”
Response: We thank the reviewer for the suggestion. The sentence is re-worded as follows: “Similar approach using scRNA-Seq ….”
Comment #2-7: Results - what was the number of sequencing reads obtained?
Response: Thanks for your helpful comment. We add this information as follows: “More than 12,500 hemocytes were assessed in each treatment, in which hemocytes de-rived from naïve and immune-challenged larvae were read over 36 Gb and contained 3,339 and 5,278 transcripts per cell on average, respectively.”
Comment #2-8: The Table at the beginning of the Results section is awkwardly placed.
Response: We thank the reviewer for this helpful observation. The table (Fig 1a) is now shown together with Fig. 1b.
Comment #2-9: Fig 1b appears to depict a UMAP of the combined data from both naive and immune-challenged hemocytes. Please explain the rationale for depicting them in this way. It’s virtually identical to the separate depictions of UMAPs in Fig 2a. Why not just number the clusters in Fig 2a and remove Fig 1b?
Response: We thank the reviewer for this thoughtful comment. Figure 1b provide an overview of the entire hemocyte landscape, integrating both naïve and immune-challenged cells to highlight the global structure of the dataset and the shared clustering framework used throughout the study. This combined UMAP serves as a reference map for all subsequent analyses, allowing readers to visualize the full diversity of hemocyte populations in a single panel before examining condition-specific distributions. In contrast, Figure 2a focuses on separate visualizations by treatment, which allow the reader to appreciate changes in cluster abundance and distribution between naïve and immune-challenged conditions. While the overall structure appears similar, these separate depictions emphasize biological differences rather than establishing the baseline clustering framework.
Comment #2-10: Fig 3 is not very informative and does not contribute much toward reader understanding of the data. Please consider re-making these graphs, so they can more effectively communicate the differences between naive and immune-challenged groups.
Response: Fig 3 is to illustrate the representative genes expressed in each cluster. It is further strengthened with Tables S3 and S4. With this information, we can characterize the cluster type in the physiological functions.
Comment #2-11: Figure 4 What do the different colors mean (white, dark blue, red, light blue)? Please define them in the figure legend. The pathway diagrams are helpful. What is the purpose of the locus numbers in those schematics?
Response: We thank the reviewer for this helpful comment. We add following explanation to the figure caption: “Heatmaps show expression patterns in naïve or immune challenge with different colors: white bar indicates no expression in either naïve or immune-challenged (“Challenge”) larvae, light blue bar indicates expression in naive but not in Challenge, red bar indicates expression in Challenge but not in naïve, and dark blue bar indicates expression in both naive and Challenge.”
Comment #2-12: Ln 358 and following: It’s very difficult to follow the line of reasoning presented by the authors, because the heatmaps in multi-part Fig 4 are not explained. I am unable to properly evaluate the significance of the findings presented in Fig 4 until this has been cleared up.
Response: We thank the reviewer for this helpful comment.
To explain the differential expression levels, we add the clear definitions of the heatmap color scheme to the Figure 4 legend. “White indicates no expression in either naïve or immune-challenged (“Challenge”) larvae; light blue indicates expression in naive but not in Challenge; red indicates expression in Challenge but not in naive; and dark blue indicates expression in both naive and Challenge.”
Comment #2-13: Ln 468 and following: Figure 5 – a) What are the units for the heatmap?
Response: Thank you for your helpful comment. Figure 5a changed and unite (average log2 fold change) added to both figure and the legend.
Comment #2-14: Fig 6 numbers are too small to read. Please reorganize this figure, so the reader can see the cluster numbers in the UMAP.
Response: Thanks for your helpful comment. Figure 6c has been updated with the larger font size.
Comment #2-15: Table S2- It would be helpful if Table S2 listed the cell types.
Response: Thanks for the valuable comment. We have added the cell types to each cluster.
Comment #2-16: Fig 7 is low resolution. Because of this, it is impossible to evaluate the data. Please improve resolution and resubmit.
Response: Thanks for your comment. We have updated the figure with a higher quality.

Reviewer 3 Report
Comments and Suggestions for Authors
Hemocytes are central to insect immunity, but the roles of different functional groups (types) of hemocytes in immune responses are not sufficiently characterized. Previous classifications were mainly based on morphology (four morphology-based groups) and on the expression of a limited number of marker genes.
This study provides novel insights into hemocyte functions, differentiation and gene expression in response to immune challenge by employing cutting-edge single-cell RNA sequencing (scRNA-seq) to investigate gene expression in individual hemocytes. The researchers used larvae of the moth Spodoptera exigua, which were injected either with heat-killed E. coli in PBS buffer (“Challenged”) or with PBS buffer alone (“Naive”).
By analyzing gene expression in approximately 12 thousands hemocytes per treatment group (with approximately 3 to 5 thousands transcripts detected per cell), the study showed that hemocytes could be clustered into 24 groups according to their gene expression patterns (principal component analysis, Figures 1b and 2a). Immune challenge resulted in changes in gene expression patterns and shifts in cell numbers among several hemocyte clusters, - for example, a dramatic increase in Clusters 12 and 18 and a reduction in Clusters 0 and 13 (Figure 2).
The study identified differentially expressed immune-related genes in key immune pathways for each of the 24 hemocyte clusters and presented a clear summary of these findings in Figure 4. These 24 clusters were subsequently assigned to both “classical” and novel hemocyte types. Furthermore, based on differential gene expression combined with morphological data and validation of marker gene expression, the study suggested that these 24 clusters represent distinct functional groups of hemocytes.
The paper is information-rich and provides a wealth of novel data. The experimental design and analyses are clear and appropriate, and the findings are well summarized and effectively presented. I have only minor suggestions:
- Figure 2a: Indicate the cluster numbers to which the arrows are pointing.
- Figure 6a: Extend the table by adding a column showing the functional group (among the seven: active protein synthesis, apoptosis, melanization, modulating cell shape, AMP production, calcium homeostasis, cell repair) to which each of the 24 clusters can be assigned (Lines 330–333).
Author Response
Comment #3-1: Hemocytes are central to insect immunity, but the roles of different functional groups (types) of hemocytes in immune responses are not sufficiently characterized. Previous classifications were mainly based on morphology (four morphology-based groups) and on the expression of a limited number of marker genes. This study provides novel insights into hemocyte functions, differentiation and gene expression in response to immune challenge by employing cutting-edge single-cell RNA sequencing (scRNA-seq) to investigate gene expression in individual hemocytes. The researchers used larvae of the moth Spodoptera exigua, which were injected either with heat-killed E. coli in PBS buffer (“Challenged”) or with PBS buffer alone (“Naive”). By analyzing gene expression in approximately 12 thousands hemocytes per treatment group (with approximately 3 to 5 thousands transcripts detected per cell), the study showed that hemocytes could be clustered into 24 groups according to their gene expression patterns (principal component analysis, Figures 1b and 2a). Immune challenge resulted in changes in gene expression patterns and shifts in cell numbers among several hemocyte clusters, - for example, a dramatic increase in Clusters 12 and 18 and a reduction in Clusters 0 and 13 (Figure 2). The study identified differentially expressed immune-related genes in key immune pathways for each of the 24 hemocyte clusters and presented a clear summary of these findings in Figure 4. These 24 clusters were subsequently assigned to both “classical” and novel hemocyte types. Furthermore, based on differential gene expression combined with morphological data and validation of marker gene expression, the study suggested that these 24 clusters represent distinct functional groups of hemocytes. The paper is information-rich and provides a wealth of novel data. The experimental design and analyses are clear and appropriate, and the findings are well summarized and effectively presented. I have only minor suggestions:
Response: Thank you for the constructive feedback.
Comment #3-2: Figure 2a: Indicate the cluster numbers to which the arrows are pointing.
Response: Thanks for your comment. We changed the figure.
Comment #3-3: Figure 6a: Extend the table by adding a column showing the functional group (among the seven: active protein synthesis, apoptosis, melanization, modulating cell shape, AMP production, calcium homeostasis, cell repair) to which each of the 24 clusters can be assigned (Lines 330–333).
Response: We add a new column to indicate the functional category.

Round 2
Reviewer 1 Report
Comments and Suggestions for Authors
The previous responses did not adequately address my initial comments, and the revisions have not sufficiently improved the readability or clarity of the figures. Below are my specific concerns:
1) Figure 1a: The authors added UMI per cell to Figure 1a, but the median number of genes per cell, as requested in my initial comment, was not included. The statement in their response, “the median number of genes per cell can be easily understood from this number by dividing by the total cell number” is incorrect. The median gene count per cell is not equivalent to the “total annotated genes” and cannot be derived in this manner.
2) Cluster Marker Table: These tables have been completed with L2FC and p-value compared to the one seen previously. It is still mentionning only the top three markers instead of the whole list of markers, it is lacking the percentage of cell of the cluster expressing the marker as well as the percentage of cells of the whole datasets. At last, multiple markers show dismal fold enrichment (< 0.1) or p-value = 1 which means that they are not strong markers of the cluster. This can modify significantly the interpretation of the data, meaning that the cluster may not be classified as a distinct hemocyte subpopulation.
For example, gene SPEXI_LOCUS8825 (ribosomal gene) is enriched 0.046 in cluster OE1. This value is used by authors to call for high expression of ribosomal genes in this cluster, which is incorrect.
3) Pathway Focus and Heatmaps: while the authors have added references, they have not provided a justification for their focus on the Toll/IMD, oxylipin, and PGE pathways. Additionally, the heatmaps in Figure 4 remain qualitative rather than quantitative, as previously requested.
4) Fig 5, Fig 6: the images of the cells need to be displayed at higher magnification to allow for proper assessment of both morphology and FISH signals.